# Can Large Language Models be Good Path Planners? A Benchmark and Investigation on Spatial-temporal Reasoning

## Abstract

Large language models (LLMs) have achieved remarkable success across a wide spectrum of tasks; however, they still face limitations in scenarios that demand long-term planning and spatial reasoning. To facilitate this line of research, in this work, we propose a new benchmark, termed **P**ath **P**lanning from **N**atural **L**anguage (**PPNL**). Our benchmark evaluates LLMs' spatial-temporal reasoning by formulating "path planning" tasks that require an LLM to navigate to target locations while avoiding obstacles and adhering to constraints. Leveraging this benchmark, we systematically investigate LLMs including GPT-4 via different few-shot prompting methodologies and BART and T5 of various sizes via fine-tuning. Our experimental results show the promise of few-shot GPT-4 in spatial reasoning, when it is prompted to reason and act interleavedly, although it still fails to make long-term temporal reasoning. In contrast, while fine-tuned LLMs achieved impressive results on in-distribution reasoning tasks, they struggled to generalize to larger environments or environments with more obstacles.

## 1 Introduction

Large language models (LLMs) (Brown et al., 2020; OpenAI, 2023; Raffel et al., 2020; Lewis et al., 2020) have astounded the world with their linguistic and reasoning capabilities, sparking vigorous debates among researchers about their role in advancing Artificial General Intelligence (AGI) (Bubeck et al., 2023; Marcus et al., 2023; Butlin et al., 2023). In this work, we particularly look into LLMs' capabilities in performing *spatial-temporal reasoning*. Spatial-temporal reasoning is a fundamental aspect of human intelligence. Humans possess the ability to seamlessly integrate spatial information from their surroundings into their decision-making processes, enabling them to make informed choices and take appropriate actions based on their spatial awareness. LLMs, trained solely on textual data, have been criticized by many for not possessing this skill (Agrawal, 2023; Chen et al., 2023; Bubeck et al., 2023).

To facilitate this investigation, in this work, we propose a new benchmark, termed **P**ath **P**lanning from **N**atural **L**anguage (**PPNL**), which analyzes if LLMs can perform end-to-end path planning in a grid environment. To succeed in the task, an LLM needs to comprehend the grid environment informed via natural language, and navigate to the target locations while avoiding obstacles and adhering to any constraints; as such, it demands both spatial and long-term, temporal reasoning capabilities from the LLM. Despite their similar focus of evaluating LLMs' reasoning skills in a grounded environment, existing benchmarks either require very little long-term planning (Ruis et al., 2020; Wu et al., 2021b), or additionally need modeling other modalities, which introduces potential confounding factors and makes the evaluation of LLMs hard to control and interpret (Shridhar et al., 2020; Wu et al., 2021a; Ruis et al., 2020; Wu et al., 2021b). On the other hand, while a number of works have concluded that LLMs are not appropriate for end-to-end planning (Chen et al., 2023; Valmeekam et al., 2022), their explorations were not based on the state-of-the-art LLMs (e.g., GPT-4) and prompting techniques. Our work thus fills the gap by both providing a controllable spatial-temporal reasoning benchmark and investigating how the most advanced LLMs perform on it.

Leveraging **PPNL**, we compare a set of LLMs, including both GPT-4 via few-shot prompting and the fine-tuned BART (Lewis et al., 2020) and T5 (Raffel et al., 2020) models. In particular, we ex-

perimented with four prompting approaches to augment GPT-4, including (1) the naive few-shot promptings with various numbers of task demonstrations, (2) a novel action-and-effect prompt, which guides an LLM in long-term planning by prompting it to keep track of its location change as an effect of its action, (3) a Chain-of-Thought (CoT) prompt adapted from Wei et al. (2022), which requires the LLM to reason about its strategy before taking actions, and (4) the adapted Re-Act (Yao et al., 2023) prompt, with which the LLM can interleavedly reason, act, and observe the local environment, so as to adjust its plan and correct a mistaken trajectory.

Our experimental results suggest a strong capability of GPT-4 in spatial reasoning, when it is prompted effectively. Specifically, the action-and-effect prompting leads to 21.5% improvement in success rate than the naive few-shot prompting, implying that guiding LLMs to reason about the situated knowledge can help them make better immediate decisions. Prompting GPT-4 to reason step-by-step (i.e., CoT) further enhances its performance by 3%, whereas additionally allowing the model to perceive the environment signals (i.e., ReAct) yields the best success rate, 96.1%. However, we also note that this success is mostly limited to relatively "local" reasoning; for example, the ReAct-prompted GPT-4 is shown to plan only a few steps ahead and have to frequently adjust its plan based on the environment signals. In summary, this indicates that even the state-of-the-art GPT-4 still falls short in long-term, temporal reasoning. In contrast, while the fine-tuned LLMs can achieve a close-to-perfect success rate when planning in environments similar to where they were trained (i.e., in-distribution environments), they suffer from the lack of generalization and cannot perform well in environments with different grid sizes or different number of obstacles. We will release our benchmark, along with the source code to controllably synthesize the grid environment, as well as our LLM implementation, to the public for future advancement.

## 2 PPNL: A Benchmark on Spatial-Temporal Reasoning of LLMs

Our benchmark, **PPNL**, assesses LLMs' spatial-temporal reasoning abilities by evaluating them on the task of "path planning". In Section 2.1, we introduce the task formulation and settings. The procedure for generating the benchmark dataset will be elaborated in Section 2.2, followed by metrics used to evaluate LLMs in Section 2.3.

### 2.1 The Path Planning Task

**Task Formulation:** Formally, we describe an environment by a set of $k$ obstacles $\mathbb{O} = \{O_1, O_2, ..., O_k\}$ placed on an $N \times N$ grid and a constraint $C$. Each path-planning task can then be formulated as follows: given an initial location $P_0$ and a set of $l$ goal locations $\mathbb{P} = \{P_1, ..., P_l\}$, the task for the LLM agent is to perform a list of actions $\mathbb{A} = (A_1, ..., A_t)$, such that it can successfully navigate from $P_0$ to all the goal locations while avoiding the obstacles $\mathbb{O}$ and adhering to the constraint $C$. To keep the evaluation simple, focused, and controllable, **PPNL** synthesizes $N \times N$ grid world environments. Depending on the specific task setting, the action space is slightly different, as we introduce below.

**Settings:** Our benchmark includes two task settings:

- **Single-Goal Path Planning**: The task is to visit a single goal location, i.e., $l = 1$. This is the most basic setting; however, it could still be challenging depending on the size of the grid environment as well as the contained obstacles. In this setting, the action space includes four choices, i.e., Up, Down, Left, and Right.
- **Multi-Goal Path Planning**: The task is to reach multiple goal locations, which can be used to assess the scalability of an LLM in planning. In this case, we further distinguish two different types of settings: (1) **No Constraints**, where the task is simply to visit all goal locations, and (2) **Constrained Ordering**, where a set of goal locations has to be visited before others. Because of the task complexity, there is a potential that the LLM-based navigator needs to pass a certain location without visiting it. Therefore, in addition to the four actions as in the single-goal setting, the multi-goal planning includes an action "Inspect", such that a location is considered visited only when an Inspect action takes place.

**LLM Performance Benchmarking:** To evaluate how well an LLM can plan the path, we provide it with the verbalized environment (including the locations of the obstacles) as well as the task description (including its initial and goal locations and the constraint, if any). The LLM is then prompted

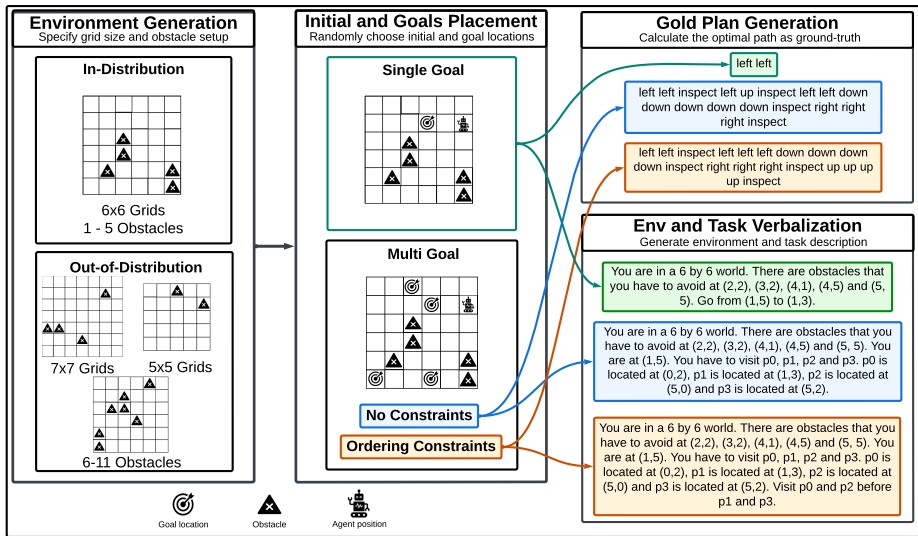

Figure 1: Overview of the data generation process.

Table 1: Statistics of **PPNL**. A breakdown of the number of environments per obstacle setup is presented in Table 6 in Appendix A.

| | # of Envs | # of Instances | |
| --- | --- | --- | --- |
| | | Single Goal | Multi Goal (No Constr./with Constr.) |
| **# In-distribution (6x6 Grids, 1-5 Obstacles)** | **836** | **25,080** | **83,600 (41,800/41,800)** |
| *# Train* | 668 | 16,032 | 53,440 (26,720/26,720) |
| *# Dev* | 668 | 2,004 | 6,680 (3,340/3,340) |
| *# Test (Unseen Placement)* | 668 | 2,004 | 6,680 (3,340/3,340) |
| *# Test (Unseen Environment)* | 168 | 5,040 | 16,800 (8,400/8,400) |
| **# Out-of-distribution (OOD)** | **400** | **12,000** | **40,000 (20,000/20,000)** |
| *# 7×7 Grids* | 125 | 3,750 | 12,500 (6,250/6,250) |
| *# 5×5 Grids* | 125 | 3,750 | 12,500 (6,250/6,250) |
| *# 6-11 Obstacles* | 150 | 4,500 | 15,000 (7,500/7,500) |

to produce a sequence of actions, aiming to accomplish the specified task in the specified environment. For each setting (i.e., single-goal or multi-goal), our benchmark covers both *in-distribution* and *out-of-distribution (OOD)* evaluation:

- **In-Distribution Generalization**, where we evaluate if an LLM can plan well in environments similar to what it has seen during the training time (i.e., fine-tuning for smaller LLMs or in-context learning for GPT-4). This includes test samples whose environments have the same grid setup and matching obstacle counts, but are different in the specific obstacle and initial/goal location placements. Therefore, they bear similar but *unseen environments*.
- **Grid-Size OOD Generalization**, where we assess how well an LLM performs in grid environments of different sizes from their training data.
- **Obstacle-Count OOD Generalization**, where we evaluate the ability of an LLM to navigate in 6×6 environments containing more obstacles than the environments used for training.

## 2.2 DATA GENERATION PROCESS

Figure 1 provides an overview of the procedure of generating **PPNL**, whose statistics are shown in Table 1. We present the details in Appendix A and summarize the process below: **(1) Environment Generation**: We consider a 6×6 grid as the in-distribution environment setup. We sampled 836 grids with random 1-5 obstacle placements as the in-distribution set, among which 668 are used for LLM development (fine-tuning or supplying the few-shot GPT-4) and the remaining 168 for unseen-environment generalization evaluation. For OOD evaluation, we sampled 25 environments for each number of obstacles, for each setting. Making up a total of 125 5×5 and 125 7×7 environments for grid-size OOD, and 150 including 6-11 obstacles for obstacle-count OOD. **(2) Initial and Goal Locations Placement**: We choose random placements for the initial and goal locations within each environment. For single-goal, 30 placements per environment are sampled, resulting in

25,080 in-distribution task instances and 12,000 OOD; for multi-goal, we considered the number of goals ranging between 2 and 6, and synthesized 10 random placements for each, resulting in 83,600 task instances for in-distribution experiments and 40,000 for OOD evaluation. We note that some obstacle placements may divide the grids into disconnected components, and the agent and goal(s) may fall on different components rendering the goals "unreachable". We keep these cases in our dataset and evaluate whether the LLMs are able to identify them. **(3) Environment and Task Verbalization**: All environment and task specifications are presented in natural language to an LLM. To this end, we devised templates to verbalize each task instance. **(4) Ground-Truth Plan Generation**: In order to generate the ground truth paths, we use the A* algorithm (Hart et al., 1968) for the single-goal setting, and address the multi-goal setting as an instance of the Traveling Salesman Problem (TSP).

## 2.3 EVALUATION METRICS

We assess an LLM's path planning performance with the following metrics: (1) **Success Rate (%)**: the percentage of predicted paths successfully navigating from the initial location to the goal(s) while satisfying the constraint (if any); (2) **Optimal Rate (%)**: the percentage of successful paths that are also optimal (i.e., taking minimal numbers of actions); (3) **Exact Match Accuracy (%)**: the percentage of optimal paths that are identical to the calculated ground-truth plan; (4) **Feasible Rate (%)**: the percentage of predicted paths that remain within the grid's boundaries and avoid obstacles, regardless of reaching the goal(s) or not; (5) **Distance to Goal(s)**: the minimum number of actions that are required for the LLM agent to navigate from its final location to fulfill the original task. This metric is only computed for paths that are feasible but never reach the goal. All metrics, except Distance to Goal(s), are the larger the better. Finally, we note that cases with unreachable goals are not counted with the aforementioned metrics, as there does not exist a valid plan for them. Instead, we introduce a new metric, (6) **Unreachable Accuracy (%)**, indicating how often an LLM can identify unreachable goals correctly.

## 3 METHODOLOGIES

We evaluate the spatial-temporal reasoning capabilities of a set of LLMs, including both GPT-4 (OpenAI, 2023) prompted in various ways (Figure 3) and the fine-tuned BART (Lewis et al., 2020) and T5 (Raffel et al., 2020), as introduced below. For simplicity, we first describe each method based on a single-goal planning setting, and then explain its extension to the multi-goal task. For each of the prompting methods listed below (except for naive prompting), we use 7 shots of demonstrations, consisting of one demonstration from an environment for each number of obstacles (1-5), as well as two demonstrations in the "unreachable goals" setting.

**Naive Prompting:** The first approach we consider is to directly prompt the LLM with the ground-truth action sequences. Prior work has shown that including different amounts of such task demonstrations can yield different performances (Cao et al., 2020; Razeghi et al., 2022). Therefore, we explored few-shot demonstrations ranging between 5 and 15.

**Action-and-Effect Prompting:** To succeed in the path planning task, an LLM is required to plan for the long term. Nonetheless, language models have been found to face difficulty in long trajectory planning (Valmeekam et al., 2022; Chen et al., 2023). Intuitively, if the LLM can keep track of its location changes, then the long-term planning demand can be decomposed to making only short-term decisions. Drawing from this intuition, we propose the action-and-effect prompting method, which prompts an LLM to reason about the effect of its action.

**Chain of Thought Prompting:** The Chain-of-Thought (CoT) approach of Wei et al. (2022), which prompts an LLM to reason "step by step" in a chain, has been shown to be effective for tasks that require multiple steps of reasoning and decision-making. Therefore, we adapt it to the task of path planning. Specifically, we apply this strategy to guide an LLM in deciding the starting action. In this process, the LLM is prompted to reason about (1) the relative direction of the goal to its current location, and (2) similarly, the relative direction of the obstacles, as well as the action to avoid them.

**ReAct Prompting:** Finally, we emulate how an agent can interleavedly perceive environment signals, reason about its plan, and take an action, following the approach of ReAct (Yao et al., 2023). Specifically, we provide two types of signals to the LLM: (1) informing the LLM when it has suc-

Table 2: Single-goal performance of each LLM on the in-distribution, unseen-environment test set. Results within parentheses are on environments containing 4 or more obstacles. The best results over **all approaches** or in-context learning only are marked.

|  | Success (↑) | Optimal (↑) | Exact Match (↑) | Feasible (↑) | Distance (↓) | Unreachable Acc (↑) |
|---|---|---|---|---|---|---|
| **In-Context Learning** | | | | | | |
| Naive few-shot (5) | 0.518(0.459) | 0.518(0.459) | 0.389(0.345) | 0.658(0.569) | 2.45(2.55) | 0.000(0.000) |
| Naive few-shot (10) | 0.497(0.440) | 0.497(0.440) | 0.371(0.339) | 0.641(0.566) | 2.73(2.62) | 0.000(0.000) |
| Naive few-shot (15) | 0.542(0.471) | 0.542(0.471) | 0.407(0.339) | 0.662(0.591) | 2.30(2.42) | 0.000(0.000) |
| Action-Effect | 0.757(0.670) | 0.757(0.670) | 0.471(0.440) | 0.757(0.679) | 0.00(0.00) | 0.000(0.000) |
| CoT | 0.787(0.710) | 0.787(0.710) | 0.526(0.484) | 0.808(0.723) | 2.43(2.00) | 0.000(0.000) |
| ReAct | 0.961(0.943) | 0.871(0.818) | 0.551(0.512) | **1.000(1.000)** | 5.26 (2.00) | 0.000(0.000) |
| **Fine-tuned Models** | | | | | | |
| BART-base | 0.808(0.813) | 0.799(0.801) | 0.743(0.751) | 0.948(0.933) | 1.29(1.28) | **0.588(0.500)** |
| BART-large | 0.941(0.917) | 0.934(0.911) | 0.913(0.884) | 0.958(0.937) | 1.24(1.21) | **0.588(0.500)** |
| T5-base | **0.979**(0.965) | **0.978**(0.962) | **0.970**(0.966) | 0.981(0.966) | 1.00(1.00) | 0.412(0.285) |
| T5-large | 0.977(0.961) | 0.976(0.959) | 0.970(0.947) | 0.981(0.965) | 1.02(1.02) | 0.529(0.428) |

cessfully reached the goal, and (2) reminding the LLM when it is one step ahead of any obstacle, as well as providing its current location. Particularly for the latter type, it allows the LLM to prevent potential failures and correct its ineffective plan. It is important to note that both signals are "local"; that is, the agent will only observe its surroundings and will not be provided with a global view. Therefore, to succeed in the path planning task, it is still required to perform long-term spatial-temporal reasoning. Considering the prohibitive cost of prompting an LLM iteratively, in experiments, we allow a ReAct LLM to incorporate environment signals and adjust its plan for at most three trials.

**Prompting in Multi-Goal Planning:** LLMs have been shown to struggle with complex and higher order reasoning tasks (Huang & Chang, 2023), as such, to aid the LLM with the multi-goal setting, we adopt a hierarchical approach, where we decompose the task into two steps: (1) prompting the models to generate an ordering in which the goal locations are to be visited; to this end, we use naive few-shot prompting with 5 exemplars and (2) following the ReAct prompting to find a path between each two subsequent goal locations, in a similar fashion to the single-goal setting.

**Fine-Tuning LLMs:** We also look at whether fine-tuning relatively smaller LLMs can endow them with spatial-temporal reasoning capabilities. Specifically, we experimented with BART (Lewis et al., 2020) and T5 (Raffel et al., 2020) with two sizes (base and large). The fine-tuned LLMs plan by directly generating the action sequence $\mathbb{A}$ given the natural-language environment and task description. For unreachable goals, the fine-tuned LLMs were trained to predict "Goal not reachable".

All fine-tuned LLMs were evaluated on the full test sets. However, due to the prohibitively high cost of GPT-4, all few-shot prompting-based methods were evaluated on a small sample of each test set. Specifically, for each test set, we randomly select 2 task instances in each generated environment. This results in totally 336 task instances for in-distribution, unseen environment evaluation, 250 for $7 \times 7$ grid and 250 for $5 \times 5$ grid size evaluation, and 300 for 6-11 obstacles. Although they are not the full sets, their sizes have been comparable to many LLM benchmark datasets (e.g., task sets in BIG-bench (2023)). For a precise comparison, we also provide the evaluation results of fine-tuned LLMs on the same sampled test sets in Appendix D. Implementation details can be found in Appendix B.

## 4 SINGLE-GOAL PATH PLANNING RESULTS

### 4.1 IN-DISTRIBUTION EXPERIMENTS

The performance achieved on the in-distribution, unseen-environment test set is presented in Table 2. Because of the challenge of path planning with multiple obstacles, we also present each LLM's performance on the instances that contain 4 or more obstacles. In this subsection, we discuss the different observations drawn from this table. Examples can be found in Figure 2.

**Increasing the number of few-shot exemplars aids GPT-4 in achieving the goal but does not assist in obstacle avoidance**. The performance of GPT-4's with naive prompting falls short. Enhancing the quantity of few-shot examples yields improvements, particularly when increasing the number of few-shot exemplars from 10 to 15. With 10 exemplars, the model often struggles to devise a plan that reaches the goal. In contrast, when prompted with 15 exemplars, failures primarily result from obstacles obstructing the predicted path. As shown in the first example from Figure 2, the path

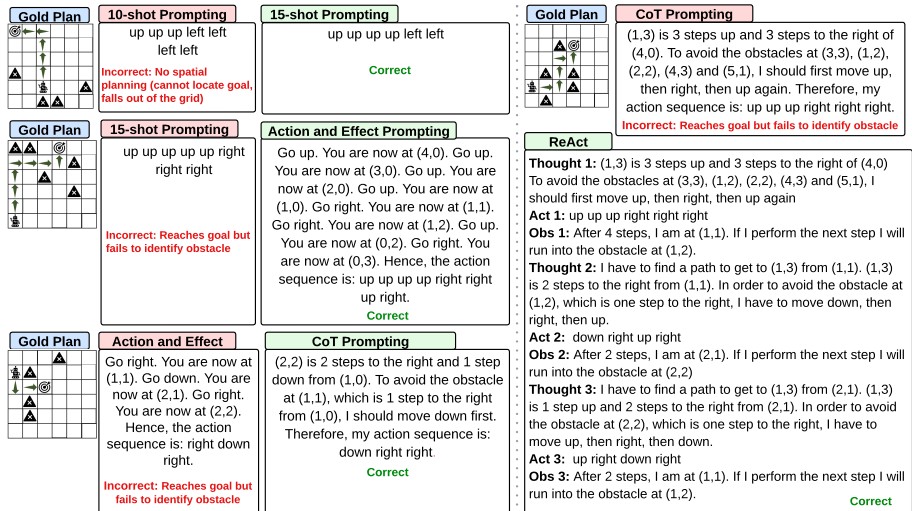

Figure 2: Overview of the most common errors produced by each prompting method for few-shot GPT-4. We show the shortcomings of each method and how other methods improve on that aspect.

predicted by using 10-shot prompting does not reach the goal and leaves the grid completely: this indicates that using this prompting method, the model is unable to learn the representation of the grid it has to navigate. Prompting with 15 examples improves in this regard. However, it still failed to avoid obstacles in many cases, as illustrated in the second example.

**Situated spatial information helps the model to make more informed decisions.** The action-and-effect prompting shows significant improvement in success rate compared with naive prompting. This suggests that the model can generate better plans when it remains cognizant of its current location and does not need to plan multiple steps ahead. In other words, the model can decide to take an action that avoids an obstacle immediately after realizing that it is about to encounter one. However, this method continued to face challenges in recognizing the presence of an obstacle when dealing with scenarios that include multiple obstacles. This can be told from the 0.00 Distance metric of Action-Effect, which indicates that, as long as the LLM does not run into obstacles or go out of boundaries, it can successfully reach the goal. As we observed, all failure cases produced by this method were due to being blocked by obstacles.

**Reasoning aids obstacle avoidance but falters for longer plans with numerous obstacles.** The CoT approach offered significant gains in performance. This indicates that GPT-4 may indeed possess a level of spatial reasoning capabilities and the ability to plan based on spatial information. For instance, it could easily generate a valid path for the test case failed by action-and-effect prompting, as illustrated in Figure 2. Nevertheless, when faced with examples involving more obstacles and cases where the goals are farther away, this technique also seemed to fail. This is apparent in the 7% performance drop faced by this approach on environments involving four or more obstacles, as well as its failure to generalize the 6-11 OOD test set.

**Feedback from the environment can help with planning.** The shortcomings of CoT are overcome when GPT-4 can additionally observe the environment signals (i.e., prompted with ReAct). Executing the valid portion of the plan and continuously improving the path by explicitly observing the effect of an action sequence through feedback from the environment, allows the model to generate one portion of the path at a time, removing the need for long-term planning capabilities for this task. This approach achieves an impressive 96.1% success rate, and also enjoys the smallest gap (2%) between the overall performance and the performance on the subset containing 4 or more obstacles. This points to the fact that GPT-4 does not struggle with spatial reasoning for our task but the difficulty comes from the temporal aspect. This hypothesis is further supported by its worst Distance metric, which indicates that, on average, the failed cases for ReAct are 5.26 steps away from the goal. For many such cases, the LLM was not able to find the correct path after three trials; hence, we expect this performance to increase by allowing more trials. Intriguingly, we found that

Table 3: Single-goal performance of each LLM on OOD evaluation sets. The best results over **all approaches** or in-context learning only are marked. The generated 7×7 grid instances do not include unreachable cases and hence show the "-" mark.

| | Success (↑) | Optimal (↑) | Exact Match (↑) | Feasible (↑) | Distance (↓) | Unreachable Acc (↑) |
|---|---|---|---|---|---|---|
| **5×5 Grid Environments** | | | | | | |
| **In-Context Learning** | | | | | | |
| CoT | 0.763 | 0.759 | 0.494 | 0.787 | 2.16 | 0.00 |
| ReAct | 0.932 | 0.819 | 0.518 | **1.000** | 4.18 | 0.00 |
| **Fine-tuned Models** | | | | | | |
| BART-base | 0.905 | 0.903 | 0.873 | 0.959 | 1.11 | 0.00 |
| BART-large | 0.964 | 0.964 | 0.955 | 0.967 | 1.07 | 0.27 |
| T5-base | **0.969** | **0.967** | **0.963** | **0.971** | 1.09 | 0.25 |
| T5-large | 0.965 | 0.965 | 0.962 | 0.969 | 1.01 | **0.32** |
| **7×7 Grid Environments** | | | | | | |
| **In-Context Learning** | | | | | | |
| CoT | 0.836 | 0.836 | 0.528 | 0.872 | 3.20 | - |
| ReAct | **0.984** | **0.871** | 0.548 | **1.000** | 3.44 | - |
| **Fine-tuned Models** | | | | | | - |
| BART-base | 0.430 | 0.422 | 0.379 | 0.869 | 1.67 | - |
| BART-large | 0.586 | 0.583 | **0.560** | 0.931 | 1.65 | - |
| T5-base | 0.548 | 0.541 | 0.524 | 0.882 | 1.57 | - |
| T5-large | 0.543 | 0.535 | 0.516 | 0.897 | 1.29 | - |
| **6-11 Obstacles** | | | | | | |
| **In-Context Learning** | | | | | | |
| CoT | 0.544 | 0.541 | 0.417 | 0.556 | 2.66 | 0.00 |
| ReAct | 0.830 | 0.729 | 0.695 | **1.0** | 7.29 | 0.20 |
| **Fine-tuned Models** | | | | | | |
| BART-base | 0.304 | 0.302 | 0.291 | 0.444 | 1.77 | **0.30** |
| BART-large | 0.369 | 0.362 | 0.345 | 0.524 | 1.54 | 0.09 |
| T5-base | **0.857** | **0.855** | **0.839** | **0.856** | 1.03 | 0.09 |
| T5-large | 0.838 | 0.835 | 0.821 | 0.854 | 1.09 | 0.15 |

all prompting methods predicted the optimal for every successful case. However, because of these iterative trials, ReAct shows an optimal rate which is 9% lower than its success rate.

**Fine-tuning can be an effective approach.** As we can see from Table 2, BART-large, as well as both varieties of T5 achieve a good performance, with T5-base achieving the highest success rate of around 98%. Fine-tuned LLMs' performance appears to be impressive at first glance. When the models failed, they still retained a close distance to the goal. In most cases, their plans are optimal and the same as the calculated ground truth. However, these models are known to fail when generalizing to out-of-distribution examples; we evaluate this hypothesis in Section 4.2.

**All models fail at identifying unreachable goals.** All models achieved very low scores on unreachable accuracy. This points to the fact that these models fail to reason about the "big picture" when it comes to spatial concepts. However, we also note that these instances were underrepresented in our data (e.g., with only 2 unreachable examples in unseen environments test set used for single-goal prompting), hence, no definitive conclusion can be reached. Future work can build on our benchmark to further explore this hypothesis.

## 4.2 OUT-OF-DISTRIBUTION EXPERIMENTS

We present results on the three OOD evaluation sets in Table 3. For few-shot GPT-4, we only experimented with CoT and ReAct as they gave the most promising performance in the in-distribution evaluation and thus showed the potential of generalizing to OOD.

**Fine-tuned models can generalize to simpler grids but not to more complex ones.** We observed that the fine-tuned LLMs were able to generalize well to the 5x5 environments. We hypothesize that this is because 5x5 environments are inherently part of the 6x6 environments encountered in training. For the 7x7 environments, however, all of the fine-tuned models failed to perform well. This highlights OOD generalization as an important limitation of fine-tuned models.

**GPT-4 can generalize to larger grids but struggles when more obstacles are involved.** GPT-4 prompted with ReAct achieved near-perfect performance in grid-size generalization; this goes along with the hypothesis that larger auto-regressive models such as GPT-4 are better at OOD generalization when compared to their smaller fine-tuned counterparts. However, while GPT-4, prompted with ReAct, achieved a success rate that was much higher than the BART models on the 6-11 obstacles,

Table 4: Multi-goal performance of each LLM on the in-distribution, unseen-environment test set. Results within parentheses are on environments containing 4 or more obstacles.

| | Success (↑) | Optimal (↑) | Exact Match (↑) | Feasible (↑) | Distance (↓) | Unreachable Acc (↑) |
|---|---|---|---|---|---|---|
| **No Constraints** | | | | | | |
| T5-base | 0.948(0.923) | 0.845(0.818) | 0.659(0.633) | 0.974(0.951) | 7.56(9.02) | 0.880(0.855) |
| ReAct | 0.893(0.857) | 0.389(0.286) | 0.132(0.129) | 1.000(1.000) | 4.06(4.91) | 0.000(0.000) |
| **With Constraints** | | | | | | |
| T5-base | 0.947(0.910) | 0.862(0.828) | 0.732(0.691) | 0.961(0.933) | 11.0(11.7) | 0.866(0.851) |
| ReAct | 0.840(0.712) | 0.385(0.350) | 0.115(0.113) | 1.000(1.000) | 5.07(5.47) | 0.000(0.000) |

Table 5: Multi-goal performance of T5-base on OOD evaluation sets.

| | Success (↑) | Optimal (↑) | Exact Match (↑) | Feasible (↑) | Distance (↓) | Unreachable Acc (↑) |
|---|---|---|---|---|---|---|
| **5×5 Grid Environments** | | | | | | |
| w/ Constraints | 0.936 | 0.855 | 0.726 | 0.945 | 5.68 | 0.267 |
| No Constraints | 0.932 | 0.835 | 0.659 | 0.951 | 3.29 | 0.261 |
| **7×7 Grid Environments** | | | | | | |
| w/ Constraints | 0.205 | 0.189 | 0.163 | 0.754 | 7.47 | 0.000 |
| No Constraints | 0.207 | 0.187 | 0.150 | 0.758 | 7.45 | 0.000 |
| **6-11 Obstacles** | | | | | | |
| w/ Constraints | 0.735 | 0.667 | 0.551 | 0.761 | 11.2 | 0.241 |
| No Constraints | 0.776 | 0.693 | 0.539 | 0.814 | 8.19 | 0.219 |

its performance was very close to that of the T5 models. On the other hand, CoT's performance was subpar when tasked with navigating environments consisting of more obstacles than the ones seen in the few shot exemplars.

## 5 MULTI-GOAL PATH PLANNING RESULTS

Next, we look at how the best-performing models on the single-goal setting scale up to handle multiple goals. Namely, We evaluate end-to-end planning using T5-base, as well as how ReAct performs when multiple goals are to be visited. We show the in-distribution evaluation results in Table 4. For OOD evaluation, we only experimented with fine-tuned T5-base (Table 5), as we have already discussed ReAct's ability to generalize to OOD settings in the single-goal setting. This skill is independent of the number of goals involved.

**Fine-tuned models can scale up to multi-goal path planning scenarios.** While the exact match accuracy of T5-base is much lower compared to the single-goal setting, the value of the success rate is very close. The lower exact match values, particularly for the case with no constraints, are due to the fact that with this task, a larger number of paths can be taken to reach the goal. The exact match accuracy as well as the optimal rate are much higher for the fine-tuned model when compared to GPT-4. This indicates that these models are more capable of inferring the algorithm used, and would be more suitable to serve as optimizers. Furthermore, we notice a considerable increase in unreachable accuracy within this setting. This may be due to the fact that the dataset used for this task was considerably larger and contained more instances involving unreachable goals.

As far as OOD generalization is concerned, we notice a trend similar to that of the single-goal OOD generalization. T5-base is still able to perform well on the 5x5 environments; however, it failed catasrophically when faced with larger grids, while also experiencing slighter performance drops on the 6-11 obstacles set.

**ReAct is not as successful when multiple goals are involved.** As we can see from Table 4, ReAct fails to achieve the same success from the single goal setting in a multi-goal context. The vast majority of failures occur with tasks involving environments involving 4 or 5 obstacles and scenarios involving six goals. We hypethosize that this is due to the accumulation of the error rate produced by the ReAct method on the single-goal path planning task.

## 6 DISCUSSION AND ANALYSIS

We performed further analyses with detailed results in Appendix D.

**Does the performance depend on the number of obstacles?** More obstacles make the planning task more challenging, as we expect (Figure 4), and GPT-4 (w/ CoT) particularly struggles in such cases, indicating that GPT-4 may still have difficulty in complex spatial reasoning.

**Is it harder to reach distant goals?** Both fine-tuned and few-shot LLMs suffer from performance drops when reaching the goals requires a long distance (Figure 5). This is a particularly greater challenge for few-shot GPT-4, suggesting its weakness in long-term, temporal planning.

**Is GPT-4 able to find the optimal path?** An interesting observation in our single-goal in-distribution evaluation (Table 2) is that, whenever an LLM (except ReAct) succeeds, it succeeds with the optimal plan, although we did not explicitly prompt it for this optimality in the task instruction (Figure 3). However, this is not true in the multi-goal experiments (Table 4). To understand this phenomenon, we conducted an additional experiment contrasting the GPT-4 performance when it is explicitly prompted for optimality or not, aiming to see if it has the capability to do optimization. From the results (Table 10) we notice that explicit prompting does improve the rate of optimal plans. However, in both cases, the rate is still low (around 50%). How much an LLM can perform optimization has attracted much interest recently (Yang et al., 2023), and we believe our benchmark can be a great resource to facilitate this line of exploration.

**Cost considerations:** Table 13 provides the average cost per sample for all approaches. ReAct prompting has an average cost far exceeding other methods, as it repeatedly queries the API to adjust the path. This high cost renders it impractical for applications involving long distance navigation. Hence, it is crucial to develop less costly approaches.

## 7 RELATED WORK

**Spatial Reasoning:** The spatial reasoning capabilities of LLMs are debated. A number of studies (Abdou et al., 2021; Ilharco et al., 2021; Patel & Pavlick, 2022; Bubeck et al., 2023) have demonstrated that LLMs can learn spatial concepts from text. However, other researchers (Agrawal, 2023; Xie et al., 2023; Wu et al., 2023) highlight weaknesses in spatial reasoning and spatial planning. In our work, we assess whether LLMs can perform spatial reasoning and pair it with reasoning about change. A number of benchmarks have been proposed to evaluate neural models' ability to perform various tasks in grounded environments (Côté et al., 2018; Shridhar et al., 2020; 2021; Ruis et al., 2020; Wu et al., 2021b). However, most focus on the spatial aspect, with short-sighted temporal planning. Our work proposes a benchmark based on path planning to evaluate LLMs' ability to combine spatial and temporal reasoning.

**Path and Motion Planning:** Many works have assessed whether LLMs can serve as planners (Song et al., 2022; Huang et al., 2022a; Valmeekam et al., 2022). However, many arrive to the conclusion LLMs are unsuitable to be end-to-end planners (Liu et al., 2023; Chen et al., 2023; Xie et al., 2023; Silver et al., 2022). We perform an in-depth analysis to verify this claim. Robot motion and path planning is an exciting application for LLMs that has been explored recently (Ahn et al., 2022; Ding et al., 2023; Chen et al., 2023; Driess et al., 2023; Huang et al., 2022b). Nevertheless, applications of LLMs in the realm of robotics remain limited. Our benchmark can, hence, be particularly useful to the robotic path and motion planning community, as we provide a controlled environment for studying the capability of LLMs to serve as path planners, and their ability to generalize to novel situations, a skill that is crucial in a large number robotic applications.

## 8 CONCLUSIONS AND FUTURE WORK

In this work, we analyze the ability of LLMs to perform spatial-temporal reasoning through using a newly proposed dataset, focusing path planning. Our findings highlight LLMs' proficiency in spatial reasoning when it is guided by situated spatial information and continuous feedback from the environment they are grounded to. However, they face significant challenges when faced with scenarios requiring long-term planning and complex environments. While ReAct prompting achieved good results, this approach can be expensive and inapplicable to situations requiring long distance navigation, or finding the optimal path. Fine-tuned models were able to perform well on this task, however, their success is limited solely to the environments similar to those encountered during in training, making them inapplicable to a large variety of applications that require robustness and adaptation to unfamiliar settings. Finding approaches that can improve LLMs' ability to perform path planning efficiently and optimally can, therefore, open the door to a wide array of application of LLMs in fields such as robotics and motion planning. We believe that our analysis and the proposed task can serve as a valuable resource for future work exploring this topic.

REPRODUCIBILITY STATEMENT

The source code to reproduce all of the experiments as well as the datasets will be made publicly available. The process to generate our dataset is thoroughly described in Section 2 as well as Appendix A. We also provide implementation details and the full prompts used to run our experiments in Appendix B.

ETHICS STATEMENT

We are committed to conducting this research within the highest ethical standards. Our aim is contribute positively to the domains of artificial intelligence and robotics, with transparency and reproducibility as central components in our design. Our dataset was generated synthetically, hence, we do not have any concerns as far as data privacy and security are concerned. However, LLMs are known to embed a large number of biases in their pre-training; it is, hence, our responsibility as artificial intelligence researchers to better unpack the inner working of these models so as to address this problem. Our benchmark and analysis can be a step-stone towards understanding the reasoning capabilities of these models.

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

## A    Data Generation Process

### A.1    Samples Generation

Figure 1 provides an overview of the procedure of generating **PPNL**. This includes first generating the environment, then placing the initial and goal locations, and then synthesizing the verbalized environment and task description as well as searching for the optimal path used as ground truth in LLM learning. While one can synthesize an arbitrary number of environments and tasks owing to the synthetic nature of **PPNL**, in this work, we create a dataset totaling more than 160,000 samples, which are summarized in Table 1.

**Environment Generation:**    We first created environments for the in-distribution evaluation. Our dataset considers the $6 \times 6$ grid as the in-distribution environment. In total, we sampled 836 grids of this size with the number of obstacles ranging between 1 and 5. Among them, 80% (668 environments) are held for LLM development (e.g., fine-tuning or used to supply the few-shot prompting of GPT-4), and the remaining 20% (168 environments) are used for the "unseen environment" generalization evaluation. For OOD evaluation, we created 125 environments with a smaller ($5 \times 5$) grid size, 125 with a larger ($7 \times 7$) grid size, and 150 with more obstacles (ranging between 6 and 11). These OOD environments are synthesized with evenly distributed numbers of obstacles. Each of these test sets includes 25 environments from each number of obstacles.

**Initial and Goal Location Placement:**    Given a generated environment with certain obstacles, we then place the initial and target locations to form individual task instances. For single-goal path planning, we performed 30 random placements for each environment. This results in 25,080 single-goal task instances for in-distribution experiments and 12,000 for OOD evaluation. For multi-goal path planning with or without constraints, we considered the number of goals ranging between 2 and 6, and synthesized 10 random placements for each, resulting in 83,600 task instances for in-distribution experiments and 40,000 for OOD evaluation. As we place the agent and the goal locations randomly, it happens that goals in some of the generated environments are not reachable, e.g., when the agent's initial location or any goal location is surrounded by obstacles. Instead of eliminating those cases, we keep them to benchmark if an LLM can be aware of such "unreachable goals" situations. We present a breakdown of the percentage of these cases for all subsets in our dataset in Table 7.

**Environment and Task Verbalization:**    We prompt an LLM to perform the path planning task by providing both the environment and the task command in natural language. This is implemented with a number of templates. The environment verbalization starts with a description of the grid size (e.g., "You are in a 6 by 6 world"), which is followed by a description of the obstacle locations. We then describe the task by specifying the initial and target location(s) as well as any constraints. Details of the templates can be found in Table 8.

**Ground-Truth Plan Generation:**    To generate the ground-truth plan for each task instance, we formulate the path planning problem as a graph search problem. We used the A* pathfinding algorithm (Hart et al., 1968) to find the optimal solutions for the single-goal planning, and combined this approach with a Traveling Salesman Problem (TSP) solver to calculate solutions for the multi-goal setting. A more detailed description of the problem formulations is included in Section A.2.

**Data Splits for LLM Fine-Tuning:**    Finally, we note the split for LLM fine-tuning. That is, for both single-goal and multi-goal settings, we split the task instances created from the 668 environments into training, dev, and test sets, following an 80%, 10%, and 10% basis. In particular, the test set contains examples from the same environments as in training time but with different agent and goal(s) placements, which we refer to as an "unseen placement" test set. However, all three sets are mainly used for LLM fine-tuning and will not be discussed for LLM benchmarking.

### A.2    Generating the Plans

**Graph Construction:**    We consider the grids as an undirected graph, where each location is a node connected to the four neighboring locations, if there exists no obstacle in these locations. More formally, for a grid $W$ and a set of obstacles $\mathbb{O}$ we construct an undirected graph $G$, where each node $u_{xy} = (x, y) \notin \mathbb{O}$ is connected to all nodes $v$ such that $v \in \{(x-1, y), (x+1, y), (x, y-1), (x, y+1)\}$ and $v \notin \mathbb{O}$

Table 6: Environments statistics breakdown by the number of obstacles.

| Subset | Overall | Number of Obstacles | | | | |
|---|---|---|---|---|---|---|
| | | 1 | 2 | 3 | 4 | 5 |
| **# In-distribution (6x6 Grids)** | 836 | 36 | 200 | 200 | 200 | 200 |
| *# Training and Dev* | 668 | 28 | 160 | 160 | 160 | 160 |
| *# Dev* | 668 | 28 | 160 | 160 | 160 | 160 |
| *# Test (Unseen Environments)* | 168 | 8 | 40 | 40 | 40 | 40 |
| **# Out-of-distribution** | 400 | 50 | 50 | 50 | 50 | 50 |
| *# 7x7 Grids* | 125 | 25 | 25 | 25 | 25 | 25 |
| *# 5x5 Grids* | 125 | 25 | 25 | 25 | 25 | 25 |
| *# 6-11 Obstacles* | 150 | 25 per obstacle setting | | | | |

Table 7: Percentage of scenarios involving unreachable goals.

| | Single Goal | Multi-goal |
|---|---|---|
| **# In-distribution (6x6 Grids, 1-5 Obstacles)** | | |
| *# Train* | 0.41% | 0.89% |
| *# Dev* | 0.55% | 1.02% |
| *# Test (Unseen Placement)* | 0.40% | 0.78% |
| *# Test (Unseen Environment)* | 0.34% | 0.89% |
| **# Out-of-distribution** | | |
| *# 7x7 Grids* | 0.00% | 0.10% |
| *# 5x5 Grids* | 0.55% | 2.82% |
| *# 6-11 Obstacles* | 15.37% | 29.07% |

**A\* Search:** In order to generate the optimal paths, which were used as ground truths, we rely on the A\* search algorithm (Hart et al., 1968); a popular method for path-finding across a variety of computer science tasks and domains. We use the Manhattan distance to the goal as a heuristic in all settings. For the single-goal setting, the algorithm was run from the initial location, with the aim of finding a set of valid locations to reach the desired goal location.

**Traveling Salesman Modeling:** For the multi-goal setting, we approach the problem by finding the optimal path between each pair of cities using A\*, and running a Traveling Salesman Problem (TSP) solver to find an optimal ordering of cities. We use the Gurobi Optimizer (Gurobi Optimization, LLC, 2023) to provide an optimal ordering of locations to visit. Note that, the optimal path between two locations may pass over certain goal coordinates. For the purposes of this work, we don't consider the consider the location lying on the path as visited, until it is reached using the order provided by the TSP solver. In order to formulate the problem mathematically, we assume the start location is an additional location to be visited. We define a decision variable $X_{ij}$ for each pair of locations $(u_i, u_j)$, where $X_{ij} = 1$ if $u_j$ is to be visited after $u_i$ and $X_{ij} = 0$ otherwise. We also define a variable $D_{ij}$, representing the shortest distance between $u_i$ and $u_j$; this is computed using the A\* algorithm. The objective function to be minimized is, hence, defined as $\sum_i \sum_j D_{ij} X_{ij}$. The following constraints should then be satisfied for all settings.

Table 8: Natural language templates for verbalizing the environment and task descriptions.

| Setting | Template |
|---|---|
| World Description | You are in a $\{N\}$ by $\{N\}$ world. |
| | There obstacles that you have to avoid at $\{obstacles\}$. |
| Enumerating Goals | $p\{i\}$ is located at $\{(x_i, y_i)\}$ |
| Single goal | Go from $\{(x_{init}, y_{init})\}$ to $\{(x_{goal}, y_{goal})\}$. |
| Multi Goal | Visit the following locations: $\{locations\}$. |
| Constrained Ordering | Visit $\{locations_a\}$ before $\{locations_b\}$ |
| Initial Location | You are at $\{(x_{init}, y_{init})\}$ |

1. **Each location should be visited exactly once**: $\sum_j x_{ij} = 1 \forall i \in |P| + 1, i \neq j$
2. **Each location should be exited exactly once**: $\sum_j x_{ji} = 1 \forall i \in |P| + 1, i \neq j$
3. **Subtour Elimination:** $\sum_j T_i - T_j + |P| * X_{ij} \leq |P|, \forall 2 \leq i, j \leq |P|$
4. **Binary Values:** $\forall i \neq j \; X_{ij} \in \{0, 1\}$

Additionally, we define an order variable $T_i$: the position of location $u_i$ in the tour, in order satisfy the ordering constraints. The following constraints are then added to model the different settings.

1. **Initial location is visited first:** We always assign $u_0$ to the initial location, hence we model this mathematically as: $T_0 = 0$
2. **Ordering Constraints:** To model the contraint that location $u_i$ should be visited before $u_j$, we use the constraint $T_i < T_j$

# B  IMPLEMENTATION DETAILS

All in-context learning experiments of GPT-4 were performed with the version "gpt-4"[1] by the time of this submission, with the temperature set to 0. The fine-tuned LLMs were implemented using the Hugging Face Transformers library (Wolf et al., 2020) and were trained on four A100:40gb NVIDIA GPUs, with a per-device batch size of 16. The training data for fine-tuning LLMs were derived from the "#Train" split in Table 1, and the "#Dev" split was used for early stopping. We use the special tokenization technique proposed by Rai et al. (2023). In Figure 3, we present one example for each prompting method. Their specific scripts are listed in the following subsections.

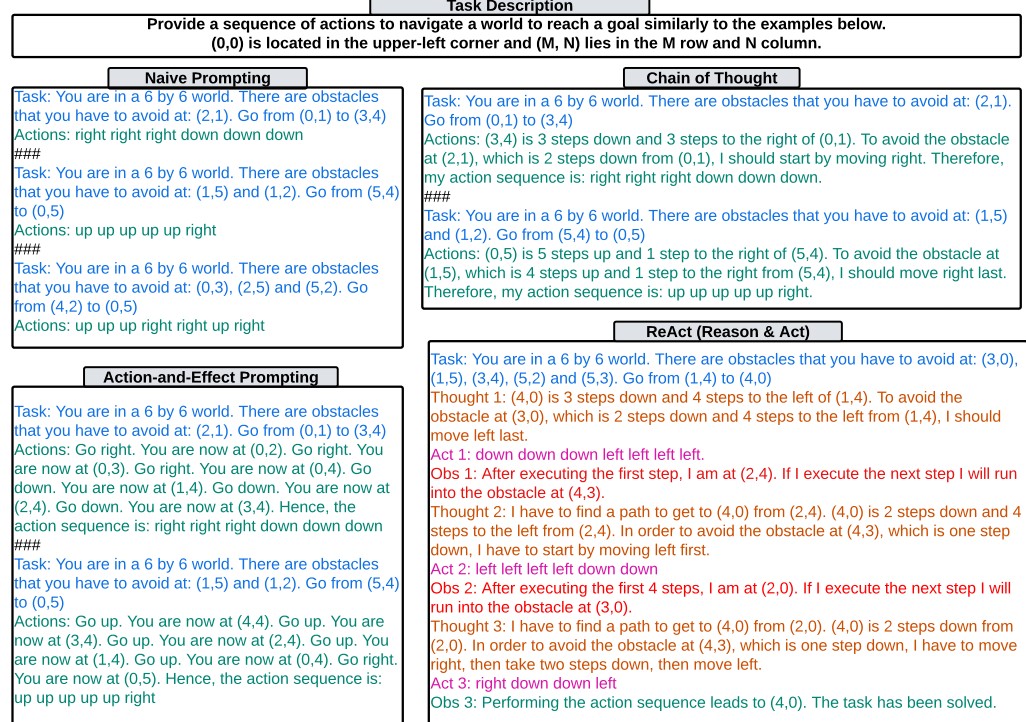

Figure 3: Overview of prompting methods for few-shot GPT-4.

## B.1  NAIVE PROMPTS

**5-shot Prompt**

Provide a sequence of actions to navigate a world to reach a goal similarly to the examples below. (0,0) is located in the upper-left corner and (M, N) lies in the M row and N column.
###
**Task**: You are in a 6 by 6 world. There are obstacles that you have to avoid at: (2,1). Go from (0,1) to (3,4)
**Actions**: right right right down down down
###
**Task**: You are in a 6 by 6 world. There are obstacles that you have to avoid at: (1,5) and (1,2). Go from (5,4) to (0,5)
**Actions**: up up up up up right
###
**Task**: You are in a 6 by 6 world. There are obstacles that you have to avoid at: (0,3), (2,5) and (5,2). Go from (4,2) to (0,5)
**Actions**: up up up right right up right
###
**Task**: You are in a 6 by 6 world. There are obstacles that you have to avoid at: (3,5), (4,2), (3,3) and (0,4). Go from (1,5) to (3,1)
**Actions**: left left left left down down
###
**Task**: You are in a 6 by 6 world. There are obstacles that you have to avoid at: (2,5), (5,2), (0,4), (1,4) and (0,1). Go from (4,2) to (1,2)
**Actions**: up up up

---

[1] https://platform.openai.com/docs/models/gpt-4

**10-shot Prompt**

Provide a sequence of actions to navigate a world to reach a goal similarly to the examples below. (0,0) is located in the upper-left corner and (M, N) lies in the M row and N column.
###
**Task**: You are in a 6 by 6 world. There are obstacles that you have to avoid at: (2,1). Go from (0,1) to (3,4)
**Actions**: right right right down down down
###
**Task**: You are in a 6 by 6 world. There are obstacles that you have to avoid at: (0,4). Go from (5,4) to (2,4)
Actions: up up up
###
**Task**: You are in a 6 by 6 world. There are obstacles that you have to avoid at: (0,4) and (1,5). Go from (0,5) to (1,1)
**Actions**: Goal not reachable
###
**Task**: You are in a 6 by 6 world. There are obstacles that you have to avoid at: (1,5) and (5,0). Go from (5,5) to (0,1)
**Actions**: up up up left up up left left left
###
**Task**: You are in a 6 by 6 world. There are obstacles that you have to avoid at: (0,3), (2,5) and (5,2). Go from (4,2) to (0,5)
**Actions**: up up up right right up right
###
**Task**: You are in a 6 by 6 world. There are obstacles that you have to avoid at: (0,3), (2,1) and (4,2). Go from (1,5) to (0,5)
**Actions**: up
###
**Task**: You are in a 6 by 6 world. There are obstacles that you have to avoid at: (2,4), (4,4), (5,3) and (4,5). Go from (0,4) to (5,5)
**Actions**: Goal not reachable
###
**Task**: You are in a 6 by 6 world. There are obstacles that you have to avoid at: (5,1), (4,4), (1,4) and (1,5). Go from (5,5) to (3,0)
**Actions**: up up left left left left left
###
**Task**: You are in a 6 by 6 world. There are obstacles that you have to avoid at: (2,5), (5,2), (0,4), (1,4) and (0,1). Go from (4,2) to (1,2)
**Actions**: up up up
###
**Task**: You are in a 6 by 6 world. There are obstacles that you have to avoid at: (0,5), (5,0), (5,4), (0,0) and (5,3). Go from (5,2) to (2,4)
**Actions**: up up up right right

---

**15-shot Prompt**

---

Provide a sequence of actions to navigate a world to reach a goal similarly to the examples below. (0,0) is located in the upper-left corner and (M, N) lies in the M row and N column. ###

**Task**: You are in a 6 by 6 world. There are obstacles that you have to avoid at: (2,1). Go from (0,1) to (3,4)
**Actions**: right right right down down down
###
**Task**: You are in a 6 by 6 world. There are obstacles that you have to avoid at: (0,4). Go from (5,4) to (2,4)
**Actions**: up up up
###
**Task**: You are in a 6 by 6 world. There are obstacles that you have to avoid at: (5,3). Go from (2,4) to (4,3)
**Actions**: left down down
###
**Task**: You are in a 6 by 6 world. There are obstacles that you have to avoid at: (0,4) and (1,5). Go from (0,5) to (1,1)
**Actions**: Goal not reachable
###
**Task**: You are in a 6 by 6 world. There are obstacles that you have to avoid at: (1,5) and (5,0). Go from (5,5) to (0,1)
**Actions**: up up up left up up left left left
###
**Task**: You are in a 6 by 6 world. There are obstacles that you have to avoid at: (4,3) and (2,3). Go from (5,5) to (5,4)
**Actions**: left
###
**Task**: You are in a 6 by 6 world. There are obstacles that you have to avoid at: (0,3), (2,5) and (5,2). Go from (4,2) to (0,5)
**Actions**: up up up right right up right
###
**Task**: You are in a 6 by 6 world. There are obstacles that you have to avoid at: (0,3), (2,1) and (4,2). Go from (1,5) to (0,5)
**Actions**: up
###
**Task**: You are in a 6 by 6 world. There are obstacles that you have to avoid at: (1,0), (0,0) and (1,3). Go from (3,2) to (3,1)
**Actions**: left
###
**Task**: You are in a 6 by 6 world. There are obstacles that you have to avoid at: (3,5), (4,2), (3,3) and (0,4). Go from (1,5) to (3,1)
**Actions**: left left left left down down
###
**Task**: You are in a 6 by 6 world. There are obstacles that you have to avoid at: (5,1), (4,4), (1,4) and (1,5). Go from (5,5) to (3,0)
**Actions**: up up left left left left left
###
**Task**: You are in a 6 by 6 world. There are obstacles that you have to avoid at: (2,4), (4,4), (5,3) and (4,5). Go from (0,4) to (5,5)
**Actions**: Goal not reachable.
###
**Task**: You are in a 6 by 6 world. There are obstacles that you have to avoid at: (2,5), (5,2), (0,4), (1,4) and (0,1). Go from (4,2) to (1,2)
**Actions**: up up up
###
**Task**: You are in a 6 by 6 world. There are obstacles that you have to avoid at: (0,5), (5,0), (5,4), (0,0) and (5,3). Go from (5,2) to (2,4)
**Actions**: up up up right right
###
**Task**: You are in a 6 by 6 world. There are obstacles that you have to avoid at: (3,0), (2,3), (1,2), (2,5) and (0,0). Go from (4,3) to (5,4)
**Actions**: right down

## B.2    ACTIONS AND EFFECTS PROMPT

---

**Actions and Effects Prompt**

Provide a sequence of actions to navigate a world to reach a goal similarly to the examples below. (0,0) is located in the upper-left corner and (M, N) lies in the M row and N column.
###
**Task**: You are in a 6 by 6 world. There are obstacles that you have to avoid at: (2,1). Go from (0,1) to (3,4)
**Actions**: Go right. You are now at (0,2). Go right. You are now at (0,3). Go right. You are now at (0,4). Go down. You are now at (1,4). Go down. You are now at (2,4). Go down. You are now at (3,4). Hence, the action sequence is: right right right down down down
###
**Task**: You are in a 6 by 6 world. There are obstacles that you have to avoid at: (1,5) and (1,2). Go from (5,4) to (0,5)
**Actions**: Go up. You are now at (4,4). Go up. You are now at (3,4). Go up. You are now at (2,4). Go up. You are now at (1,4). Go up. You are now at (0,4). Go right. You are now at (0,5). Hence, the action sequence is: up up up up up right
###
**Task**: You are in a 6 by 6 world. There are obstacles that you have to avoid at: (0,3), (2,5) and (5,2). Go from (4,2) to (0,5)
**Actions**: Go up. You are now at (3,2). Go up. You are now at (2,2). Go up. You are now at (1,2). Go right. You are now at (1,3). Go right. You are now at (1,4). Go up. You are now at (0,4). Go right. You are now at (0,5). Hence, the action sequence is: up up up right right up right
###
**Task**: You are in a 6 by 6 world. There are obstacles that you have to avoid at: (3,5), (4,2), (3,3) and (0,4). Go from (1,5) to (3,1)
**Actions**: Go left. You are now at (1,4). Go left. You are now at (1,3). Go left. You are now at (1,2). Go left. You are now at (1,1). Go down. You are now at (2,1). Go down. You are now at (3,1). Hence, the action sequence is: left left left left down down
###
**Task**: You are in a 6 by 6 world. There are obstacles that you have to avoid at: (2,5), (5,2), (0,4), (1,4) and (0,1). Go from (4,2) to (1,2)
**Actions**: Go up. You are now at (3,2). Go up. You are now at (2,2). Go up. You are now at (1,2). Hence, the action sequence is: up up up
###
**Task**: You are in a 6 by 6 world. There are obstacles that you have to avoid at: (0,4) and (1,5). Go from (0,5) to (1,1)
**Actions**: Goal not reachable.
###
**Task**: You are in a 6 by 6 world. There are obstacles that you have to avoid at: (2,4), (4,4), (5,3) and (4,5). Go from (0,4) to (5,5)
**Actions**: Goal not reachable.

---

## B.3 Chain of Thought Prompt

---

**CoT Prompt**

Provide a sequence of actions to navigate a world to reach a goal similarly to the examples below. (0,0) is located in the upper-left corner and (M, N) lies in the M row and N column. ###

**Task**: You are in a 6 by 6 world. There are obstacles that you have to avoid at: (2,1). Go from (0,1) to (3,4)

**Actions**: (3,4) is 3 steps down and 3 steps to the right of (0,1). To avoid the obstacle at (2,1), which is 2 steps down from (0,1), I should start by moving right. Therefore, my action sequence is: right right right down down down.
###

**Task**: You are in a 6 by 6 world. There are obstacles that you have to avoid at: (1,5) and (1,2). Go from (5,4) to (0,5)

**Actions**: (0,5) is 5 steps up and 1 step to the right of (5,4). To avoid the obstacle at (1,5), which is 4 steps up and 1 step to the right from (5,4), I should move right last. Therefore, my action sequence is: up up up up up right.
###

**Task**: You are in a 6 by 6 world. There are obstacles that you have to avoid at: (0,3), (2,5) and (5,2). Go from (4,2) to (0,5)

**Actions**: (0,5) is 4 steps up and 3 steps to the right of (4,2). I can start by going up. To avoid the obstacle at (0,3), which is 4 steps up and 1 step to the right of (4,2), I should take a right at (1,3). Therefore my action sequence is: up up up right right up right
###

**Task**: You are in a 6 by 6 world. There are obstacles that you have to avoid at: (3,5), (4,2), (3,3) and (0,4). Go from (1,5) to (3,1)

**Actions**: (3,1) is 4 steps to the left and 2 steps to the right of (1,5). No obstacles fall on this path. Therefore my action sequence is: left left left left down down.
###

**Task**: You are in a 6 by 6 world. There are obstacles that you have to avoid at: (2,5), (5,2), (0,4), (1,4) and (0,1). Go from (4,2) to (1,2)

**Actions**: (1,2) is 3 steps up from (4,2). No obstacles fall on this path. Therefore my action sequence is: up up up.
###

**Task**: You are in a 6 by 6 world. There are obstacles that you have to avoid at: (0,4) and (1,5). Go from (0,5) to (1,1)

**Actions**: (0,5) is surrounded by obstacles. Therefore, the goal is not reachable from my location.
###

**Task**: You are in a 6 by 6 world. There are obstacles that you have to avoid at: (2,4), (4,4), (5,3) and (4,5). Go from (0,4) to (5,5)

**Actions**: (5,5) is surrounded by obstacles. Therefore, the goal is not reachable from my location.

---

## B.4 REACT PROMPT

---

**ReAct Prompt**

Provide a sequence of actions to navigate a world to reach a goal similarly to the examples below. (0,0) is located in the upper-left corner and (M, N) lies in the M row and N column.
###
**Task**: You are in a 6 by 6 world. There are obstacles that you have to avoid: (2,3), (5,5) and (5,2). Go from (0,4) to (5,0)
**Thought 1**: (5,0) is 5 steps down and 4 steps to the left of (0,4). To avoid the obstacle at (2,3), which is 2 steps down and 1 step to the left from (0,4), I should move left first.
**Act 1**: left left left left down down down down down.
**Obs 1**: Performing the action sequence leads to (5,0). The task has been solved.
###
**Task**: You are in a 6 by 6 world. There are obstacles that you have to avoid: (3,0), (1,5), (3,4), (5,2) and (5,3). Go from (1,4) to (4,0)
**Thought 1**: (4,0) is 3 steps down and 4 steps to the left of (1,4). To avoid the obstacle at (3,0), which is 2 steps down and 4 steps to the left from (1,4), I should move left last.
**Act 1**: down down down left left left left.
**Obs 1**: After executing the first step, I am at (2,4). If I execute the next step I will run into the obstacle at (4,3).
**Thought 2**: I have to find a path to get to (4,0) from (2,4). (4,0) is 2 steps down and 4 steps to the left from (2,4). In order to avoid the obstacle at (4,3), which is one step down, I have to start by moving left first.
**Act 2**: left left left left down down
**Obs 2**: After executing the first 4 steps, I am at (2,0). If I execute the next step I will run into the obstacle at (3,0).
**Thought 3**: I have to find a path to get to (4,0) from (2,0). (4,0) is 2 steps down from (2,0). In order to avoid the obstacle at (4,3), which is one step down, I have to move right, then take two steps down, then move left.
**Act 3**: right down down left
**Obs 3**: Performing the action sequence leads to (4,0). The task has been solved.
###
**Task**: You are in a 6 by 6 world. There are obstacles that you have to avoid: (0,5), (3,5), (0,0), (4,5) and (4,4). Go from (1,1) to (5,0)
**Thought 1**: (5,0) is 4 steps down and 1 step to the left of (1,1). To avoid the obstacle at (0,0), which is 1 step up and 1 step to the left from (1,1), I should move down first.
**Act 1**: down down down down left.
**Obs 1**: Performing the action sequence leads to (5,0). The task has been solved.
###
**Task**: You are in a 6 by 6 world. There are obstacles that you have to avoid: (5,5), (5,0), (3,5) and (4,0). Go from (3,4) to (1,3)
**Thought 1**: (1,3) is 2 steps up and 1 step to the left of (3,4). No obstacles fall on this path.
**Act 1**: up up left
**Obs 1**: Performing the action sequence leads to (1,3). The task has been solved.
###
**Task**: You are in a 6 by 6 world. There are obstacles that you have to avoid: (1,3), (4,2), (3,3) and (1,4). Go from (3,2) to (4,4)
**Thought 1**: (4,4) is 1 step down and 2 step to the right of (3,2). To avoid the obstacle at (3,3) which is one step to the right, I have to go down first
**Act 1**: down left left
**Obs 1**: If I execute the first step I will run into the obstacle at (4,2).
**Thought 2**: I have to find a path to get to (4,4) from (3,2). (4,4) is 1 step down and 2 steps to the right of (3,2). In order to avoid the obstacle at (4,2) which is one step down, and the obstacle at (3,3) which one step to the right, I have to move up, take two steps to the right and two steps down.
**Act 2**: up right right down down
**Obs 2**: Performing the action sequence leads to (4,4). The task has been solved.
###
**Task**: You are in a 6 by 6 world. There are obstacles that you have to avoid: (0,3), (1,2), (3,5) and (0,1). Go from (0,2) to (3,1)
**Thought 1**: (3,1) is 3 steps down and 1 step to the left of (0,2). To avoid the obstacle at (1,2), which is 1 step down from (0,2), I should start by moving down.
**Act 1**: down down down left
**Obs 1**: If I execute the first step I will run into the obstacle at (1,2).
**Thought 2**: (0,2) is surrounded by obstacles. Therefore, the goal is not reachable from my location.
**Act 2**: No action
**Obs 2**: No action is to be performed. The goal is not reachable. The task has been solved.
###
**Task**: You are in a 6 by 6 world. There are obstacles that you have to avoid: (2,5), (5,2), (0,4), (1,4) and (0,1). Go from (4,1) to (1,5)
**Thought 1**: (1,5) is 3 steps up and 4 steps to the right of (4,1). To avoid the obstacle at (2,5), which is 2 steps up and 4 steps to the right from (4,1), I should move right last.
**Act 1**: up up up right right right right

**Obs 1**: After executing the first 5 steps, I am at (1,3). If I execute the next step I will run into the obstacle at (1,4).
**Thought 2**: (1,5) is surrounded by obstacles. Therefore, the goal is not reachable from my location.
**Act 2**: No action
**Obs 2**: No action is to be performed. The goal is not reachable. The task has been solved.

---

## B.5 Ordering Prompts

---

**Prompting for a valid ordering**

Provide a plan to navigate a world to reach all the goals while satisfying any constraints similarly to the examples below. (0,0) is located in the upper-left corner and (M, N) lies in the M row and N column.
###
**Task**: You are in a 6 by 6 world. There are obstacles that you have to avoid at: (5,2), (2,3) and (5,0). You are at (0,2). You have to visit p0, p1, p2, p3 and p4. p0 is located at (3,5), p1 is located at (5,4), p2 is located at (2,4), p3 is located at (3,2) and p4 is located at (4,4). Visit p1 and p3 before p0, p2 and p4.
**Order**: p3, p1, p4, p2, p0
###
**Task**: You are in a 6 by 6 world. There are obstacles that you have to avoid at: (2,5), (0,2) and (4,5). You are at (4,2). You have to visit p0, p1, p2, p3 and p4. p0 is located at (0,1), p1 is located at (2,2), p2 is located at (1,2), p3 is located at (5,3) and p4 is located at (5,5).
**Order**: p3, p4, p1, p2, p0
###
**Task**: You are in a 6 by 6 world. There are obstacles that you have to avoid at: (1,0), (3,3) and (1,1). You are at (3,5). You have to visit p0, p1, p2, p3, p4 and p5. p0 is located at (1,3), p1 is located at (0,4), p2 is located at (4,0), p3 is located at (2,4), p4 is located at (5,0) and p5 is located at (5,4). Visit p4, p3 and p2 before p0, p1 and p5.
**Order**: p3, p2, p4, p5, p0, p1
###
**Task**: You are in a 6 by 6 world. There are obstacles that you have to avoid at: (0,1). You are at (4,0). You have to visit p0, p1, p2, p3 and p4. p0 is located at (3,2), p1 is located at (1,1), p2 is located at (2,2), p3 is located at (0,4) and p4 is located at (1,5). Visit p2 and p3 before p0, p1 and p4
**Order**: p2, p3, p4, p1, p0
###
**Task**: You are in a 6 by 6 world. There are obstacles that you have to avoid at: (2,1). You are at (5,3). You have to visit p0 and p1. p0 is located at (2,5) and p1 is located at (2,2). Visit p1 before p0
**Order**: p1, p0

---

## Prompting for an optimal ordering

Provide an optimal plan to navigate a world to reach all the goals while satisfying any constraints similarly to the examples below. (0,0) is located in the upper-left corner and (M, N) lies in the M row and N column. A path is optimal if it satisfies the constraints using the minimum number of actions

###

**Task**: You are in a 6 by 6 world. There are obstacles that you have to avoid at: (5,2), (2,3) and (5,0). You are at (0,2). You have to visit p0, p1, p2, p3 and p4. p0 is located at (3,5), p1 is located at (5,4), p2 is located at (2,4), p3 is located at (3,2) and p4 is located at (4,4). Visit p1 and p3 before p0, p2 and p4.

**Order**: The optimal plan is: p3, p1, p4, p2, p0

###

**Task**: You are in a 6 by 6 world. There are obstacles that you have to avoid at: (2,5), (0,2) and (4,5). You are at (4,2). You have to visit p0, p1, p2, p3 and p4. p0 is located at (0,1), p1 is located at (2,2), p2 is located at (1,2), p3 is located at (5,3) and p4 is located at (5,5).

**Order**: The optimal plan is: p3, p4, p1, p2, p0

###

**Task**: You are in a 6 by 6 world. There are obstacles that you have to avoid at: (1,0), (3,3) and (1,1). You are at (3,5). You have to visit p0, p1, p2, p3, p4 and p5. p0 is located at (1,3), p1 is located at (0,4), p2 is located at (4,0), p3 is located at (2,4), p4 is located at (5,0) and p5 is located at (5,4). Visit p4, p3 and p2 before p0, p1 and p5.

**Order**: The optimal plan is: p3, p2, p4, p5, p0, p1

###

**Task**: You are in a 6 by 6 world. There are obstacles that you have to avoid at: (0,1). You are at (4,0). You have to visit p0, p1, p2, p3 and p4. p0 is located at (3,2), p1 is located at (1,1), p2 is located at (2,2), p3 is located at (0,4) and p4 is located at (1,5). Visit p2 and p3 before p0, p1 and p4.

**Order**: The optimal plan is: p2, p3, p4, p1, p0

###

**Task**: You are in a 6 by 6 world. There are obstacles that you have to avoid at: (2,1). You are at (5,3). You have to visit p0 and p1. p0 is located at (2,5) and p1 is located at (2,2). Visit p1 before p0

**Order**: The optimal plan is: p1, p0

## C  FINE-TUNED MODELS' PERFORMANCE ON SAMPLED TEST SETS

Due to the prohibitive cost of running GPT-4, our evaluation of few-shot GPT-4 has been based on a subset of each full evaluation set. For a precise comparison, in Table 9 we summarize the results achieved by each fine-tuned LLM on the same, smaller evaluation sets as the few-shot GPT-4. We notice that the performance does not change significantly when compared to the performance on the full set. Hence, we believe that the small set is representative of the full test sets.

Table 9: Fine-tuned Models' Performance on Sampled Test Sets

| | Success (↑) | Optimal (↑) | Exact Match (↑) | Feasible (↑) | Distance (↓) | Unreachable Acc (↑) |
|---|---|---|---|---|---|---|
| **In-Distribution (Unseen Environments)** | | | | | | |
| BART-base | 0.790 | 0.775 | 0.708 | 0.952 | 1.33 | 1.0 |
| BART-large | 0.934 | 0.931 | 0.913 | 0.952 | 1.26 | 1.0 |
| T5-base | 0.982 | 0.979 | 0.973 | 0.982 | 1.00 | 1.0 |
| T5-large | 0.985 | 0.982 | 0.979 | 0.988 | 1.03 | 0.5 |
| **5×5 Grid Environments** | | | | | | |
| BART-base | 0.907 | 0.904 | 0.875 | 0.943 | 1.11 | 0.00 |
| BART-large | 0.959 | 0.959 | 0.947 | 0.959 | 1.08 | 0.00 |
| T5-base | 0.963 | 0.963 | 0.959 | 0.972 | 1.04 | 0.00 |
| T5-large | 0.965 | 0.965 | 0.962 | 0.969 | 1.01 | 0.00 |
| **7×7 Grid Environments** | | | | | | |
| BART-base | 0.501 | 0.500 | 0.448 | 0.888 | 1.82 | - |
| BART-large | 0.616 | 0.584 | 0.604 | 0.928 | 1.76 | - |
| T5-base | 0.596 | 0.588 | 0.572 | 0.912 | 1.69 | - |
| T5-large | 0.596 | 0.584 | 0.564 | 0.924 | 1.49 | - |
| **6-11 Obstacles** | | | | | | |
| BART-base | 0.347 | 0.347 | 0.343 | 0.478 | 1.67 | 0.05 |
| BART-large | 0.386 | 0.386 | 0.374 | 0.548 | 1.54 | 0.122 |
| T5-base | 0.872 | 0.860 | 0.860 | 0.872 | 1.00 | 0.122 |
| T5-large | 0.841 | 0.841 | 0.822 | 0.854 | 1.05 | 0.195 |
| **Multi Goal End-to-end Planning** | | | | | | |
| No Constraints | 0.960 | 0.849 | 0.685 | 0.974 | 6.0 | 0.0 |
| w/ Constraints | 0.954 | 0.856 | 0.706 | 0.954 | 0.0 | 0.0 |

## D  ADDITIONAL ANALYSIS

**Does the performance depend on the number of obstacles?** Figure 4 points to the fact that adding more obstacles can adversely affect performance. Nevertheless, T5-base appears to fair much better than CoT. While prompting the model to think step-by-step does indeed allow it to make better decisions, it fails when there are multiple obstacles to be avoided. This could indicate that while GPT-4 possesses a basic level of spatial reasoning, it struggles to utilize this skill for complex reasoning chains.

**Is it harder to reach distant goals?** Looking at Figure 5a, it is evident that T5's performance starts to decline as the ground truth path length increases. In other words, T5 encounters difficulties in finding solutions for reaching distant goals. A similar trend can be observed for GPT-4 when prompted with ReAct, as depicted in Figure 5b. However, it is worth noting that GPT-4 appears to face even greater challenges in achieving distant targets. This suggests that GPT-4 is less successful at long-term planning. This approach's failure is particularly apparent in terms of optimal rate, further supporting the claim that while ReAct can improve the models' ability to reach the desired destination through path correction, it still fails to improve the temporal reasoning aspect of our task, as finding the optimal path requires planning on longer horizons.

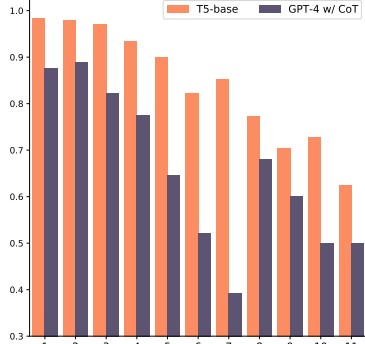

Figure 4: The success rate of T5-based and GPT-4 with CoT in the single-goal setting by the number of obstacles

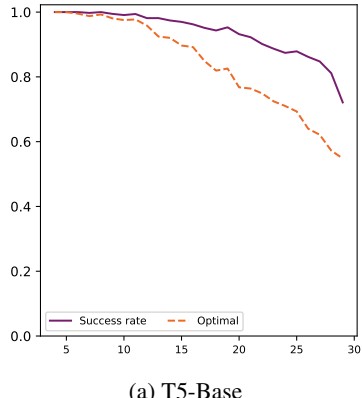 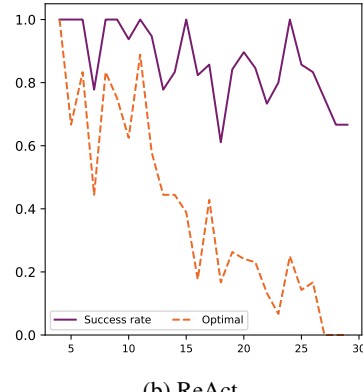

          (a) T5-Base                     (b) ReAct

Figure 5: Performance of the best-performing fine-tuned LLM (T5-base) and ReAct based on the path length of the ground-truth plan on the multi-goal unseen environments test set.

**Is GPT-4 able to find the optimal path?** An interesting observation drawn from Table 2 is that whenever any of the prompting techniques were able to find a solution without the need for correction in the single-goal setting, the solution was also optimal, despite the need for an optimal solution not being mentioned in the prompt. However, we noticed a significant drop in this metric when scaling up to multiple goals.

Table 10: Results of GPT-4 when prompted for optimal order or not.

|  | Constraints Satisfied | Optimal Ordering |
|---|---|---|
| **No Constraints** | | |
| Prompting w/ no Optimality | - | 0.47 |
| Prompting w/ Optimality | - | 0.51 |
| **With Constraints** | | |
| Prompting w/ no Optimality | 1.0 | 0.52 |
| Prompting w/ Optimality | 1.0 | 0.56 |

The ordering generated in our hierarchical approach was not always optimal. Hence, we conduct an additional experiment where we specifically prompt the models for the optimal order. We use A* search to compute the optimal distance between every two subsequent locations in the generated ordering and evaluate the optimality of the plan.

From Table 10, we notice that explicitly prompting the models for the optimal does indeed offer slight improvements on this metric. However, the overall performance is still quite low. Achieving the optimal ordering requires the model to implicitly solve an instance of TSP. Doing this requires significant temporal reasoning capabilities. Exploring ways to improve on this metric for the multi-goal setting can be an interesting avenue of research to be explored as part of future work. Recent work (Yang et al., 2023) has proposed methods to utilize LLMs as optimizers, which can be a promising avenue of research. We believe that our benchmark can be a great resource for researchers wishing to explore this idea.

**Are the fine-tuned models better at generalizing to unseen placements?** We also look at whether the fine-tuned models are better at generalizing to novel initial and goal placements within environments used for training as opposed to generalizing to unseen environments. Our results, presented in 11, show that the models achieve almost the same performance on this test; indicating that these models do not overfit to the environments they are trained on.

Table 11: Fine-tuned models' performance on the unseen placements test set

|  | Success (↑) | Optimal (↑) | Exact Match (↑) | Feasible (↑) | Distance (↓) | Unreachable Acc (↑) |
|---|---|---|---|---|---|---|
| BART-base | 0.820 | 0.807 | 0.783 | 0.945 | 1.27 | 0.625 |
| BART-large | 0.929 | 0.926 | 0.911 | 0.9525 | 1.23 | 0.875 |
| T5-base | 0.971 | 0.967 | 0.962 | 0.977 | 1.02 | 0.625 |
| T5-large | 0.973 | 0.967 | 0.973 | 0.978 | 1.01 | 1.000 |

**Does the action space matter?** Another question we ask is whether representing the task differently affects the models' performance. To explore this, we propose an alternative setting where we constrain the agent's movement to the direction it is facing. In this case, the action space comprises

three instances: turn left, turn right, and move forward. We assume the agent always starts facing south.

We evaluate the performance of the T5-base model using this representation in the single-goal setting and present the results in Table 12. With this representation, we observe a slight decrease in performance, indicating that the manner in which we present the task to the models is an important consideration. Due to cost and time limitations, we did not conduct all the experiments considered in this work using this representation. However, we have included solutions based on this representation in our benchmark, which will be made publicly available for use by other researchers who wish to further explore this question.

Table 12: T5-base performance considering agent direction

| Test Set | Exact Match | Success Rate | Distance | Feasible | Optimal |
|---|---|---|---|---|---|
| Unseen Placements | 0.967 | 0.967 | 3.02 | 0.975 | 0.965 |
| Unseen Environments | 0.966 | 0.968 | 2.11 | 0.976 | 0.967 |
| 5x5 Environments | 0.954 | 0.954 | 3.48 | 0.962 | 0.954 |
| 7x7 Environments | 0.527 | 0.546 | 0.885 | 0.893 | 0.533 |
| 6-11 Obstacles | 0.749 | 0.749 | 18.11 | 0.775 | 0.754 |

**Cost considerations:** Table 13 summarizes the average cost and inference time required to produce a sequence for each prompting technique in the single-goal experiments (except the last row). The values for the average price are estimated based on the values provided by OpenAI [2].

Table 13: Average Cost per Sample for Each Prompting Technique

| Approach | Input tokens | Output tokens | Cost ($) |
|---|---|---|---|
| Naive few-shot (5) | 418.85 | 3.91 | 0.012 |
| Naive few-shot (10) | 722.85 | 3.86 | 0.022 |
| Naive few-shot (15) | 1013.86 | 3.95 | 0.031 |
| Actions+effects | 789.85 | 58.02 | 0.027 |
| CoT | 685.86 | 54.37 | 0.024 |
| ReAct | 1028.79 | 81.555 | 0.036 |
| ReAct (multi-goal) | 3086.37 | 244.665 | 0.107 |

**Can the results scale up to more realistic environments?** Finally we look at whether the results we observed can be generalized to a more complex environment, that more closely resembles a real-world navigation scenario. We prompt GPT4-V(ision) using Naive Prompting, Action Effects Prompting and Chain-of-Thought Reasoning. The current version of GPT-4V struggled to learn from few-shot image-and-text examples, even when we tried providing the information over multiple conversational turns. As a result, all of our experiments were done in a zero-shot setting only. However, our findings from small 2D grid environments still held true here - we showed that (a) vanilla zero-shot GPT-4V just generates random action sequences, (b) augmenting GPT-4V with Action-Effect knowledge dramatically improves success rate, and (c) zero-shot GPT-4V with CoT gives similar output to GPT-4V Action-Effect due to the zero-shot limitation, but it also outlines a strategy of asking for environment feedback, further supporting the potential of ReAct.

---

[2] https://openai.com/pricing

## Planning with GPT-4V(ision): Naive Prompting

**Image:**

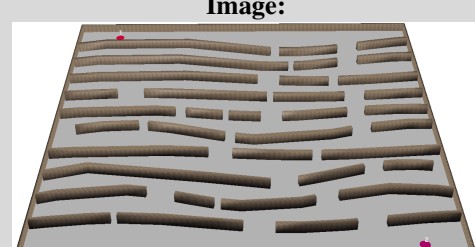

**Naive Prompting:** You are navigating the environment below, each elevated object is an obstacle that you have to avoid, Go from point 1 to point 2. Point 1 is located in the top left corner and point 2 is located in the bottom right corner. Your only available actions are (left, right, up, down). Your answer should consist of the predicted sequence solely.

**Naive Prompting Output:** down, down, down, down, down, down, right, right, down, down, left, left, down, down, right, right, down, down, left, left, down, down, right, right, down, down, left, left, down, down, right, right, down, down, left, left, down, down, right, right, down, down, left, left, down, down, right, right, down, down, left, left, down, down, right, right, down, down, left, left, down, down, right, right, down, down, left, left, down, down, right, right, down, down, left, left, down, down, right, right, down, down, left, left, down, down, right, right, down, down, left, left, down, down, right, right, down, down, left, left, down, down, right, right, down, down, left, left, down, down, right, right, down, down, left, left, down, down, right, right, down, down, left, left, down, down, right, right, down, down, left, left, down, down, right, right, down, down, left, left, down, down, right, right, down, down, left, left, down, down, right, right, down, down, left, left, down, down, right, right, down, down, left, left, down, down, right, right, down, down, left, left, down, down, right, right, down, down, left, left, down, down, right, right, down, down, left, left, down, down, right, right, down, down, left

**Remarks:** Completely random sequence. Consistent with our finding that naive prompting fails at conceptualizing the task and environment.

## Planning with GPT-4V(ision): Action-Effect Prompting

**Image:**

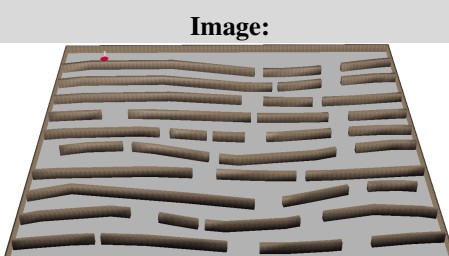

**Action and Effects Prompting:** You are navigating the environment below, each elevated object is an obstacle that you have to avoid, Go from point 1 to point 2. Your only available actions are (left, right, up, down). Find a path to go from point 1 to point 2. Point 1 is located in the top left corner and point 2 is located in the bottom right corner. Show me the predicted action sequence and what you think would happen with every action

**Action and Effects Prompting Output:** Action: Move down. Outcome: You would move closer to the first obstacle row, avoiding collision. Action: Move down. Outcome: You would be positioned just before the first horizontal obstacle, still avoiding collision. Action: Move right. Outcome: You would navigate towards the gap between the first and second obstacle in the first row. Action: Move down. Outcome: You would now be aligned with the gap between the obstacles, moving closer to the second row. Action: Move right. Outcome: You would enter the second row, moving through the gap between the first and second obstacles. Action: Move down. Outcome: You would be positioned just before the obstacles in the second row. Action: Move right, and continue this action. Outcome: You would pass through the gaps between the obstacles in the subsequent rows, moving down each row closer to point 2. Action: Once aligned with point 2, move right. Outcome: You would reach point 2 located at the bottom right corner of the environment.

**Remarks:** With this prompting method, GPT-4v is able to better reason about the environment and how to avoid obstacles, but the plan generated falls short, as it does not reach the goal.

**Planning with GPT-4V(ision): Chain-of-Thought Prompting**

**Chain of Thought Prompting:** You are navigating the environment below, each elevated object is an obstacle that you have to avoid, Go from point 1 to point 2. Point 1 is located in the top left corner and point 2 is located in the bottom right corner. Your only available actions are (left, right, up, down). Provide a sequence of actions to reach point 2 from point 1. Think step-by-step

**Chain of Thought Prompting Output:** Start at Point 1. Move right until the path is obstructed by an obstacle. When an obstacle is directly to the right, move down until you are able to move right again. Continue moving right until you encounter another obstacle. Repeat the process of moving down and then to the right until you can no longer move right because the path is either clear or because you have reached the last column before Point 2. Once in the column adjacent to Point 2, move down until you are aligned horizontally with Point 2. Move right into Point 2.

**Remarks:** The predicted approach outlines a strategy based on environment feedback (e.g. Move right until the path is obstructed by an obstacle.), decomposing the task into segments that require moving in a single direction until an obstacle is encountered; which highlights that when these methods are augmented with a method similar to ReAct to identify when an obstacle has been encountered, it can successfully solve the task. Nevertheless, the solution generated is not optimal as it requires naively reasoning locally to avoid obstacles.

**Image:**

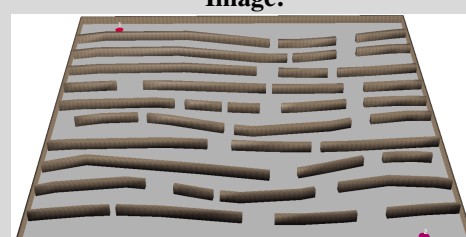

