# OpenReview forum: "Can Large Language Models be Good Path Planners? A Benchmark and Investigation on Spatial-Temporal Reasoning"
_ICLR.cc/2024/Conference — Submitted to ICLR 2024_

### Official Review · Reviewer_wB3J · 2023-10-29

**Soundness:** 3 good
**Presentation:** 3 good
**Contribution:** 1 poor
**Rating:** 5
**Confidence:** 3

**Summary:**

Paper presents a new benchmark to evaluate the capacity of spatial-temporal reasoning of LLMs. The format is path planning, where given a 2D grid map with obstacles and targets (given as text), the LLM is required to produce a path plan that connects all the targets (optionally in a given order) and avoids all the obstacles. The LLM is also asked to predict whether the current path planning problem is unsolvable. The experiments cover an extensive range of topics including different LLMs, prompting methods, using in-context learning (zero-shot) vs, fine-tuning, IID vs. OOD tasks, etc.

**Strengths:**

+The topic studied here is important, spatial-temporal reasoning is critical to more general intelligence, and as far as I know, there is not much evaluation on LLMs with a focus on this. I believe the research presented in this manuscript should be of interest to audiences not just from the LLM community, but reasoning and GOFAI as well.

+The benchmark is well designed. It is simple and straightforward but gets right to the point of spatial-temporal reasoning(maybe a bit short on the temporal part, more on that lately).

+The experiments are quite thorough, with rich numbers and details. From the LLM evaluation perspective, it covers most of the angles. I personally like the IID vs OOD part as it is less discussed before in the literature of LLM + reasoning and the results look quite promising as well.

**Weaknesses:**

I have some concerns regarding the motivation, the results, and some technical details (which will be listed in the question section):

-Can the author elaborate more on why the path planning task in this benchmark can be used to measure temporal reasoning? I understand there is a variant that requires reaching the goals in a pre-specified order, but this seems more of a spatial reasoning problem as the planned path ultimately unfolds into a 2D grid. How is this spatial-temporal? I need to admit I am not an expert it this but more of a curious reader and I am happy to learn more if there are kinds of literature/references about this.

-As the results show, most of the models (both in-context learning and fine-tuned) are able to attain 75+ accuracy some many critical metrics, ex. success rate, optimality, feasibility, etc. If this is the case, it seems that LLMs have almost nailed this task, why is this benchmark still useful for building better LLMs? Maybe the unreachable accuracy is still quite challenging, but this is only a small portion of the proposed benchmark.

-some references on LLM + planning are missing: [1-3]

[1] DEPS: https://arxiv.org/abs/2302.01560

[2] Plan4MC: https://arxiv.org/abs/2303.16563

[3] GITM: https://arxiv.org/abs/2305.17144

**Questions:**

-In table 2, there are some results on OOD evaluation with in-context learning. Can the authors clarify the exact settings of this? What are the data used to fine-tune the models, and what are the in-context examples when performing the OOD evaluations?

-What are the exact prompts used in the task? Specifically, what is the prompt for predicting whether the goal is reachable?

---

> ### Author Response · Authors · 2023-11-19
> **Responses to Reviewer wB3J**
>
> We thank the reviewer for spending time reviewing our work and acknowledging our contributions to the discussion of IID vs. OOD!
>
> Regarding “why the path planning task in this benchmark can be used to measure temporal reasoning” and the challenge of our benchmark, please refer to “General Responses to All Reviewers”.
>
> **Missing references**
> We thank the reviewer for suggesting the missing references! We apologize for the oversight and missing these references; we will make sure to add them to our updated draft.
>
> Looking into the suggested papers, they all revolve around planning in a minecraft environment. Below, we briefly discuss their connect to our work.
> [1] proposed an interactive approach for open-world planning based on LLMs allowing error correction through self-explanation. This, indeed is relevant to our work as we find substantial improvements on our benchmark by incorporating “interactive” environment feedback (i.e., the ReAct experiments).
>
> [2] relied on an approach where LLMs are used for skill planning. The proposed approach combines reinforcement learning with LLMs in order to decompose the planning task in a minecraft world to achieve long-term planning tasks. LLMs are used to build a skill graph by learning relationships between skills, while RL is used to plan over the learnt graph. This paper highlights another example of how LLMs can be useful tools to assist with long-term planning approaches.
>
> [3]  Used LLM to break down the goals and map them to structured actions for final control signals. The proposed LLM-based agent consists of an LLM Decomposer, an LLM Planner, and an LLM Interface. Similarly to [1], this work is very relevant to our work as it supports the conclusions reached through our ReAct agent.
>
> **Q1: Clarification on the OOD evaluation**
> For the training data for the fine-tuned models, we use the “#Train” subsets from Table 1, which contains 16k instances in single-goal setting and 53k instances in multi-goal setting, all under 6x6 grids and with 1-5 obstacles. We point the reviewer to the first as well as the last two paragraphs of Section 3 for implementation details and experimental design. for more details about the different settings. Similarly, for in-context learning, we draw few-shot examples from the training sets used for finetuning and evaluate on instances drawn from the OOD test sets.
>
> **Q2: Clarification on the “exact prompts used in the task”**
> The full prompts used for all in-context experiments are presented in Appendix B for reproducibility. The few shot exemplars are drawn from the training set. The prompt for predicting whether the goal is reachable is not a separate one, however, we provide few-shot exemplars that include this case, and we expect the LLMs to identify these cases based on these examples.

---

### Official Review · Reviewer_yt2L · 2023-10-31

**Soundness:** 3 good
**Presentation:** 3 good
**Contribution:** 2 fair
**Rating:** 6
**Confidence:** 3

**Summary:**

This paper aims to investigate spatial-temporal reasoning and planning capabilities of SOTA LLMs. It proposed PPNL, a benchmark contains a set of 2D grid path planning problems, and conducted various experiments examining several LLMs's capabilities in path planning in a number of settings: in-distribution, out-of-distribution with varying grid size and number of obstacles, and multi-goal long-term planning settings. The results show that with appropriate prompting technique, LLMs can reason well in relatively simple settings, but struggles when it comes to long-term temporal reasoning.

**Strengths:**

- The direction of the paper is important: temporal and spatial reasoning capabilties are indeed crucial for LLMs and ultimately AGI systems
- The experiments are well designed and conducted thoroughly
- Writing and paper presentation are polished

**Weaknesses:**

- My biggest concern is that the experiment setting is a bit too simple: it's just a set of discrete 2D grid, which is far from ideal and realistic path planning setting: high DoF, 3D space, continuous action. I understand 2D grid is a good starting point, but still, it doesn't provide sufficient value for revealing deep enough insight into LLM's limits. for example, such experiments don't shed light on how current LLMs can reason in 3D space
- This is a bit philosophical: spatial reasoning in a blind (pure language) space is, at least to me, not a well grounded request. I understand at the time of submitting, GPT-4v is not available yet, but there are also other large multimodal available, such as Bard. Maybe such experiments would be more justified if the reasoning is grounded with a vision input? After all, spatial path planning with only access to language description, even for humans, is not a very common task. I would like to see more insights on this from the authors.

**Questions:**

NA

---

> ### Author Response · Authors · 2023-11-19
> **Responses to Reviewer yt2L**
>
> We thank the reviewer for sharing the thoughtful comments on our work! We appreciate the reviewer’s acknowledgement on the importance of our research topic (i.e., temporal and spatial reasoning).
>
> We refer the reviewer to “General Responses to All Reviewers” regarding the concern about “the experiment setting is a bit too simple” and “multimodal” modeling. We welcome further comments from the reviewer and will be happy to have deeper discussions!

---

> > ### Comment · Reviewer_yt2L · 2023-11-22
> > **Reviewer Response**
> >
> > Thank you for your detailed response and additional experiments. My concerns have been partially addressed, but the simplicity of the task proposed still makes me not fully convinced by the significance of the contribution of the paper.
> > I have raised my score to a positive one.

---

### Official Review · Reviewer_w1hA · 2023-11-04

**Soundness:** 2 fair
**Presentation:** 3 good
**Contribution:** 2 fair
**Rating:** 3
**Confidence:** 3

**Summary:**

The paper examines the ability of Large Language Models (LLMs) to perform spatial-temporal reasoning with a focus on path planning. It presents findings on LLMs’ proficiency in spatial reasoning when provided with spatial information and feedback from the environment. The paper highlights challenges LLMs face in scenarios requiring long-term planning and complex environments. The research introduces ReAct prompting and fine-tuned models' performances on newly proposed datasets, emphasizing their limitations and potentials in robotic applications.

**Strengths:**

The strengths of the paper include a thorough analysis of LLMs' capabilities in spatial-temporal reasoning and path planning. The originality of the work is evident in the creation of a new dataset and the formulation of specific benchmarks for path planning. The quality of research seems robust, with significant clarity in presenting the challenges and potential of LLMs in complex tasks. The significance of the study is clear, as it informs the limitations of current models and outlines potential future work to improve LLMs' application in real-world tasks.

**Weaknesses:**

One primary concern is what this paper brings to the community. The conclusion is stated at the end of the introduction, which basically matches what we would expect from other recent papers, especially considering the toy nature of the tasks. Additionally, the prompting itself, as the "method" section, is also using existing stand "techniques" and will not fundamentally solve the spatiotemporal reasoning + generalization problem.

Another one of the concerns about the paper is that the domain studied, such as 7x7 path planning, could be considered somewhat simplistic or "toy-like." This raises questions about the extent to which the findings can be generalized to more complex, real-world scenarios. The use of small-scale environments may not adequately capture the challenges and nuances that would be present in larger, more intricate settings that LLMs might encounter in practical applications. If the benchmarking tasks do not accurately reflect the complexity of real-world tasks, it may limit the utility of the findings. To advance the field, it would be beneficial for future work to address scaling issues and test LLMs in more diverse and complex environments that better approximate actual use cases.

**Questions:**

1. How do the authors justify the use of the 7x7 path planning environment as a valid proxy for evaluating LLMs' true planning performance?
2. What are the authors' plans for testing LLMs in more complex and realistic environments to ensure the findings are scalable and applicable to real-world tasks?
3. Could the authors comment on any additional metrics or methods that might be used to evaluate planning performance in more complex scenarios?
4. How might advancements in LLMs impact the spatiotemporal reasoning capability?

---

> ### Author Response · Authors · 2023-11-19
> **Responses to Reviewer w1hA**
>
> We thank the reviewer for acknowledging the quality of our experimental design and the thoughtful comments they provided.
>
> **Clarification on the contribution of this work to the community**
> We refer the reviewer to the general response for a detailed discussion on the contributions of our benchmark compared with prior work.
>
> Here, we further clarify that our work has provided many novel insights beyond “what we would expect from other recent papers”, particularly on the less studied topic of “spatial-temporal reasoning”. For example,  Patel et. al (2022) studied whether text-only LLMs can ground spatial concepts (e.g. direction, color, etc.) to a grid world. They find that these concepts are, indeed, encoded in the LLMs. However, the conclusions have been limited to only concept understanding but not “reasoning”. Bubeck et. al (2023) evaluated LLMs ability to navigate a map of a house. They found that, while GPT-4 is not successful at exploring the whole house, it is able to accurately describe what it is exploring, despite being prompted solely through a text-based interface, highlighting LLMs' "spatial awareness" but also underscoring a shortcoming at planning; however the latter point was not explored in depth. Besides, as shown in the general response, most existing benchmarks do not support the assessment combining spatial and long-horizon temporal reasoning; as a result, not many insights can be obtained on this topic.
>
> Our work fills the gap by systematically evaluating a set of advanced LLMs in path planning, where both spatial and long-horizon temporal reasoning are needed. Our experiments were based on both fine-tuned LLMs and the state-of-the-art GPT-4, when it is augmented with the most advanced prompting methods (e.g., CoT, ReAct), which were not explored in prior work. Our experimental results showed that LLMs, with careful prompting, exhibit a certain level of spatial reasoning, but they still fall short in temporal reasoning.
>
> **Q1: How do the authors justify the use of the 7x7 path planning environment as a valid proxy & Q2: findings are scalable and applicable to real-world tasks**
>
> We refer the reviewer to the general response for the discussion about PNNL’s simplicity but that its findings can be generalized to more complex, real-world scenarios.
>
> **Q3: Could the authors comment on any additional metrics or methods that might be used to evaluate planning performance in more complex scenarios?**
>
> The metrics used in this work (e.g., success rate) can still be generalizable to more complex scenarios. Another useful metric for motion and path planning in complex environments is Clearance from obstacles (Plaku, 2017), which measures the distance of the agent to nearby obstacles. This metric allows for better evaluation of the plans as, in some cases, paths having lower clearances might be harder to execute when taking dynamics into account.
>
> **Q4: How might advancements in LLMs impact the spatiotemporal reasoning capability?**
>
> A key takeaway from our conclusion is that LLMs are not able to perform any mature level if spatial-temporal reasoning. A future research question to be explored is: “Are LLMs incapable of spatial-temporal reasoning, or do they just need to be prompted in a specific way to elicit such capability?”. Our work shows that interactions with environment feedback idea is very promising in this regard, however, it remains impractical due to a high cost and slow inference. Therefore, an idea that can be explored is whether LLMs are able to self-correct intrinsically without need for external signals; recent work has shown that this skill is not present in current LLMs (https://arxiv.org/pdf/2310.01798.pdf); indicating that they lack the ability for true reasoning; but they seem to encode enormous knowledge which allows them to exhibit what seems to be concept commonsense understanding (which can be reduced to a memorization task). Hence, developing low-cost and efficient approaches (e.g. smaller fine-tuned models trained with carefully designed objective functions) can yield significant improvements.

---

### Official Review · Reviewer_T3Yg · 2023-11-04

**Soundness:** 1 poor
**Presentation:** 2 fair
**Contribution:** 2 fair
**Rating:** 5
**Confidence:** 3

**Summary:**

This work proposes a benchmark on the ability of LLMs to perform path planning (PPNL - Path Planning from Natural Language) and analyzes several language models on the benchmark including fine tuned models. The authors claim that results on this benchmark demonstrate that LLMs perform spatial reasoning which can be systematically measured and improved upon via evaluation on the benchmark. They find that LLMs do not succeed in path planning on out-of-distribution data and long horizon examples.

**Strengths:**

The systematic construction of examples on which to test path planning is nicely presented. The writing in the paper is clear, and the evaluation on the proposed benchmark is thorough. The formulation of this paper as an investigation into the true characteristics of a property which people are actively trying to leverage in their model development is a strong direction. If the claim of demonstrating spatial temporal reasoning in LLMs was established in this work (see weaknesses), it would be an interesting and novel result.

**Weaknesses:**

It seems to me that the results here could be explained in an entirely different way.

Even though there are systematically constructed evaluations of increasing complex path planning problems across different dimensions (length, number of obstacles, etc.), I do not see how the fact that LLMs fail on the more complex tasks does not just imply that the pattern based instruction imitation of LLMs (the alternative interpretation of LLM instruction success) does not just fall apart more quickly on more difficult tasks. Imitating textual examples of instructions that these LLMs have been trained on in a pattern based way would yield significant success in providing directional information, particularly over short horizons.  Difficult tasks (long horizon, etc.) which require spatial reasoning have a lower probability of accidentally being successful with imitation-based responses.

In fact, the fine-tuning results where improvement is found in distribution and fail on out of distribution examples seems to support this alternate interpretation - not the authors’ interpretation.

This imitation based mimicry of problem solving in relation to the ability of LLMs to perform mathematical computations have been widely discussed in the past (Bubeck et al. 2023). Also, the referenced papers on spatio-temporal reasoning used on PPNL (CoT and ReAct) provide approaches to use LLMs for spatio-temporal reasoning which is fundamentally different from implying LLMs actually perform spatio-temporal reasoning.

In fact even in the spatial reasoning section of the related work, it does not appear that any prior work supports the idea that spatial reasoning can be performed by LLMs. There are 3 types of work cited by the authors which also represent my understanding of the community's view of this problem:

1. LLMs can be used via methodological automated prompting to develop spatial plans (the prompting method + LLM executes planning which has varying success in accomplishing the task),
2. LLMs have some level of spatial understanding (textual request for code to make images has significant success, etc.),
3. LLMs do not perform reasoning for math / planning / etc. problems - just quite good mimicry.

If the authors can explain and convince me that these results show LLMs actually perform reasoning over spatial information during the rebuttal, I would be willing to significantly increase my score.

**Questions:**

See weaknesses.

---

> ### Author Response · Authors · 2023-11-19
> **Responses to Reviewer T3Yg**
>
> We are grateful to the reviewer for their insightful comment on the state of research on LLM spatial reasoning!
>
> We noticed a misunderstanding on our experimental results. Therefore, we would first like to clarify our major observation. That is, based on our results, we believe that LLMs do exhibit __some__ ability for reasoning over spatial information. However, they are still __not perfectly__ solving spatial reasoning, and they have failed in long-horizon temporal reasoning.
>
> Below, we clarify how we reached the conclusion that “LLMs do exhibit __some__ ability for spatial reasoning”. We believe that the observed success of LLMs is not solely “pattern/instruction imitation”. Whether LLMs can reason or not is still a debatable question and our work does not aim to resolve the argument (instead, we provided a benchmark to facilitate it). In our humble opinion, pattern imitation happens when the same or similar patterns have been provided to the LLM, either during its pre-training or in-context learning, so that the LLM may succeed by memorizing the patterns from the data and then adapting the language to the specific test input, rather than reasoning about the test problem from scratch. However, it is important to note that the path planning task we have focused on in this work is very different from the commonly seen math reasoning tasks (e.g., GSM8k, MATH) in the following ways:
>
> First of all, path planning itself is an optimization problem. As elaborated in Appendix A.2, a path planning task essentially projects to the A* search (single-goal setting) or combining A* search with the Traveling Salesman (multi-goal setting) problems. This is very different from math problem solving where an LLM could cheat by imitating the pattern of how numbers given in the condition can be combined into an equation and eventually led to a numerical answer. Due to its optimization nature, path planning cannot be achieved with trivial pattern imitation.
>
> While math problems are often framed in daily scenarios (such that an LLM may leverage the commonsense pattern it has learned to play “shortcuts”), the path planning problems in PNNL is highly symbolic (e.g., using p0 and p1 rather than the specific city names as locations, and indicating all obstacles using x-y coordinates). So playing “shortcuts” is not very likely.
> We want to clarify that our prompts to LLMs do not include the exact test environments, and can even include completely OOD ones, so there does not exist any potential “pattern” for imitation. Notably, GPT-4 with CoT obtains stable generalization to smaller or larger grid environments (success rate of 0.787 -> 0.763 and 0.836). While it suffers from degradation in the obstacle OOD evaluation, the success rate is still decent (0.544). To further uncover its underlying mechanism, we added one experiment for 168 20x20 grid environments, each consisting of 40 obstacles. We still observed a success and optimal rates of 0.279 from GPT-4 AE and 0.351 from GPT-4 CoT. This implied that LLMs do reason about their optimization strategy in path planning, although the strategy is not very generalizable. We would like to denote a distinction between (a) whether LLMs can reason and (b) where their reasoning outcome (e.g., their path planning strategy) is generalizable to the more complicated scenarios. Intuitively, even for humans (e.g., human soldiers), when they have learned skills exclusively only in a small-size practice room, they would have difficulty when entering a real-size battlefield, but this does not imply their lack of reasoning or learning during the practice.
> Lastly, while similar math problems may have potentially been included the pre-training data of LLMs, this is much less likely to be true for path planning, because intuitively people do not commonly verbalize a symbolic task on the Internet data.
>
> As discussed in our experimental results section, LLMs still cannot “perfectly” reason about the spatial information. This again can be evidenced by the degradation of LLMs in OOD evaluation. They also fail to perform long-horizon, temporal reasoning; for example, we observed that the success of ReAct compared with CoT is attributed to the way how it receives environment feedback and decomposes a long-horizon task into multiple short-horizon ones. Another result highlighting LLMs inability for temporal planning is the low ability to find the optimal path, particularly for the multi-goal setting; We particularly probe the LLM ability to perform optimization in Appendix D, and present the results in Table 10, which clearly shows that GPT-4 falls short on this task.

---

> > ### Author Response · Authors · 2023-11-19
> > **Responses to Reviewer T3Yg -- Cont.**
> >
> > In summary, our intention of the experiments is to systematically study the state of reasoning capability of LLMs when they are augmented with advanced prompting methods, but our results do not imply mature spatial-temporal reasoning of LLMs. We apologize for any confusing language in our result discussion, but we would also like to emphasize that whether LLMs can reason or not, as the reviewer pointed out, is still an open question. Our main goal of this research is to provide a better benchmark to facilitate this exploration but not to resolve it.

---

> ### Comment · Reviewer_T3Yg · 2023-11-22
> **Reply to rebuttal**
>
> Thank you very much for the thorough rebuttal. I have thought carefully about the rebuttal points and am currently keep my score.
>
> Some remarks:
>
> - "playing shortcuts is not very likely" seems like a conjecture, and I do not know that there is evidence to back up this claim. Again, I will say that it seems to me that evidence of the solution being beyond imitation is the ability to execute the same logic on out of distribution or long horizon tasks (where the models tend to fail) since these cases cannot be imitated. In terms of "number of examples" being limited, the authors fine-tune the LLMs so the models do receive examples of this type of input/output combination or execute an explicit reasoning policy on top of the LLM .
>
> - Similar and the same are quite different. I would argue that providing fine-tuned information to an LLM of 6x6 grid path planning problems and then testing on different 6x6 grid path planning problems are "similar". I would expect a method which actually performs reasoning to fairly easily generalize to 7x7 grids. I think it is a substantially overstated analogy by the authors to make the comparison to this small degree of grid size generalization with human soldiers "learn[ing] skills exclusively only in a small-size practice room, [...] hav[ing] difficulty when entering a real-size battlefield."
>
> - Regarding the simplicity of the environment which several other reviewers brought up, I understand the point that the environment is intentionally simple to probe the capability very directly for the intended planning task. I think this is a good design choice. However, a red flag regarding this proposed benchmark is the comment of the following comment from the authors combined with the fairly high performance results of the LLMs:
>
> "A key takeaway from our conclusion is that LLMs are not able to perform any mature level if spatial-temporal reasoning. A future research question to be explored is: “Are LLMs incapable of spatial-temporal reasoning, or do they just need to be prompted in a specific way to elicit such capability?”."
>
> If LLMs are not able to perform any mature level of spatial-temporal reasoning, why should we use a benchmark to measure spatial-temporal reasoning where 90+% success metrics reported in results tables for models that cannot do this reasoning? See Table 3 in the paper.

---

> > ### Author Response · Authors · 2023-11-22
> > **Response to Reviewer T3Yg**
> >
> > We thank the reviewer for checking our response, but there still seems to be a misunderstanding on our results. **We want to clarify that none of the experimental result tables should be read separately. That is, the performance of an LLM should be understood based on its performance on all the evaluation settings (IID + OOD).** For example, we never claimed any of the fine-tuned or few-shot LLMs have “solved” spatial-temporal reasoning because of the observations that the reviewer has also noticed, i.e., they still suffer from performance degradation in OOD generalization, despite the good IID performance. To make it clear, in our next version, we will introduce a single metric that consolidates the observations from both IID and OOD, so this more fair and comprehensive evaluation would not be mistaken.
> >
> >
> > > “playing shortcuts is not very likely”
> >
> > Exact match scores can, in fact, support this claim – If the LLM was merely following the exact patterns presented in the few shot example (i.e. inferring the exact process by which those exemplars were generated), we would expect the values for the exact match metric to be almost identical to the success rate (i.e. every time the model succeeds at finding a path, it is exactly identical to the gold truth). However, we notice a significant gap between these two metrics, particularly in the multi-goal setting (see Table 4)
> >
> > > “If LLMs are not able to perform any mature level of spatial-temporal reasoning, why should we use a benchmark to measure spatial-temporal reasoning where 90+% success metrics reported in results tables for models that cannot do this reasoning? See Table 3 in the paper.”
> >
> >
> > Again, we note that none of our result tables should be read separately as they can give a comprehensive and correct interpretation of an LLM’s performance only when they are read together. Therefore, seeing a 90% success rate on a single table does not confirm any signals of true capability.
> > In addition, the reviewer seemed to misread our results. In table 3, 90%+ metrics are reported for: 1) fine-tuned models on the 5x5 grids; we explain this by the fact that 5x5 grids are inherently part of the 6x6 grids, hence these models were able to navigate them, without having to go out of distribution since all of the coordinates exist in the 6x6 grids, and 2) ReAct, which is not sensitive to the grid sizes as it solves the problem “incrementally” by decomposing it into multiple subtasks.
> >
> >
> >
> > We hope our follow-up response can resolve the reviewer’s confusion on how to read our result tables. Furthermore, as the reviewer has noticed, how well LLMs can reason requires comprehensive experiments and evaluations from different angles (e.g., IID+OOD). Our main contribution, i.e., the PNNL benchmark, is developed to contribute to such discussions. We believe our benchmark, including its potential extension in the future (e.g., environment size, geometric shapes, evaluation metrics, etc.), will inspire further discussions around this topic.

---

### Author Response · Authors · 2023-11-19
**General Responses to All Reviewers**

We thank all reviewers for the thoughtful comments on our work! In this post, we will address the common confusion on the contribution of our benchmark. In our next draft, we will make revisions to reflect all the clarifications.

**Summary**
We stress our contribution of filling the gap of missing a well-scoped, controllable benchmark for evaluating the spatial-temporal reasoning capability of LLMs. Prior benchmarks investigating spatial understanding or reasoning require either no (Patel, 2022) or only very little (Ruis, 2020; Shridhar, 2020; Qiu, 2021) temporal reasoning. Some other datasets have unnecessarily complicated the LLM evaluation by introducing other modalities (see **Regarding Reviewer yt2L’s suggestion on multimodal modeling**). Our benchmark, PNNL, aims to provide an easy-to-use and scalable framework for evaluating LLMs ability to encode spatial information and leverage spatial knowledge to solve long-horizon optimization problems. In addition, owing to its synthetic task environments, PNNL allows for high-degree controllability and scalability, whereas our findings can generalize to the more complex, real-world scenarios (see **Regarding Reviewer w1hA and yt2L’s concerns about the simplicity of our benchmark**). In the table below, we summarized the uniqueness of PNNL compared with prior work.

| Benchmark | Spatial Task Complexity | Temporal Task Complexity | Task Environment | Synthetic vs Human-annotated | Optimization Needed? | Decoupled from Vision component? |
|-|-|-|-|-|-|-|
| Patel, 2022 | Understanding | None | 2D Grid | Synthetic | No | Yes |
| Ruis, 2020 | Reasoning | Short-horizon Planning | 2D Grid | Synthetic | No | No |
| Shridhar, 2020 | Reasoning | Short-horizon planning | Embodied Environment | Human-annotated | No | No |
| Côté, 2018 | Reasoning | Short-horizon planning | Embodied Environment | Synthetic | No | Yes |
| Shridhar, 2021 | Reasoning | Short-horizon planning | Embodied Environment | Human-annotated | No | Yes |
| Valmeekam, 2022 | Reasoning | Short-horizon planning | Blocksworld | Synthetic | Yes | Yes |
| **Ours** | **Reasoning** | **Long-horizon Planning** | **2D Grid** | **Synthetic** | **Yes** | **Yes** |

All of the papers in the table above were cited in our initial submission, and we apologize for not making the distinction clearer initially. In our final PDF, we will include this table for a better view of literature.

---

> ### Author Response · Authors · 2023-11-19
> **General Responses to All Reviewers -- Cont.**
>
> **Regarding Reviewer w1hA and yt2L’s concerns about the simplicity of our benchmark and whether “the findings can be generalized to more complex, real-world scenarios”**
>
> First, we would like to note that, albeit being simple, our benchmark based on the 2D grid environments is already sufficiently challenging for LLMs at their current stage. For example, while ReAct reveals impressive performance in in-distribution settings, it still suffers from OOD generalization, and as we discussed in Section 4.1, ReAct’s success does not imply any salient temporal reasoning capability; instead, it is attributed to breaking the long-horizon task into multiple short-horizon ones with the costly iterative prompting. The fine-tuned LLMs also struggled significantly in OOD generalization. Therefore, we don’t consider the path planning task being “solved”.
>
> Second, our benchmark can be easily extended to be much more complex in the future, owing to its synthetic nature. For example, by placing the obstacles strategically, we can simulate rather complicated geometric shapes (e.g., circular mazes or battlefields with scattered blocks). Additional constraints (e.g. specifying the need for the fuel level) can also be seamlessly integrated. In addition to its scalability, the synthetic nature of our benchmark also makes it resilient to “data contamination”. That is, while ChatGPT/LLMs may continually re-train themselves from the up-to-date internet data, we can refresh the benchmark (via synthesis) and ensure that newer generations of ChatGPT/LLMs have not encountered the exact problems being tested. Furthermore, developing a complicated simulation environment can be time-consuming, while our relatively simpler (yet challenging) benchmark can greatly accelerate this development process and be easier for implementing the evaluation metrics.
>
> Finally, we emphasize that the findings that we obtained from the current relatively smaller-scale grid environments can generalize to the more complex ones. To showcase this, we have performed two additional sets of experiments.
> In the first set of experiment, we performed an OOD evaluation of GPT-4 Action-Effect and CoT on 168 synthesized environments with a much larger grid size (20x20, compared to 6x6 in IID) and more obstacles (40, compared to 5 in IID). The two LLM variants obtained a success and optimal rate of 0.279 and 0.351, respectively, indicating a consistent finding as what we observed in the relatively smaller-scale grid environments.
> In the second set of experiments, we aimed to evaluate LLMs in a 3D environment (see Appendix D in our updated draft). Because of bringing in the vision component, we replaced GPT-4 with GPT-4V(ison). However, the current GPT-4V does not seem to be able to learn from few-shot image-and-text demonstrations, even when we tried to feed the information through multiple turns of conversations. Therefore, all experiments were performed in only a zero-shot setting. However, our findings from the small-scale 2D grid environments still generalized here — we showed that (a) the naive zero-shot GPT-4V only generates random action sequences, (b) augmenting GPT-4V with Action-Effect can dramatically improve the chance to succeed, and (c) GPT-4V with zero-shot CoT gives a similar output as GPT-4V Action-Effect due to the limitation of zero shot, but it also outlines a strategy of requesting environment feedback, which implies the promise of ReAct.

---

> ### Author Response · Authors · 2023-11-19
> **General Responses to All Reviewers -- Cont.**
>
> **Regarding Reviewer yt2L’s suggestion on multimodal modeling**
>
> The main aim for our task is to look at whether text-only LLMs can plan over spatial information, not to have an end-to-end navigation agent for a realistic environment. That is, we focus on “probing” LLMs and forming better understanding on their capability scope. Prior benchmarks investigated spatial reasoning for vision-language models as a whole (Lake, 2018; Shridhar, 2020; Qiu, 2021); however, with our benchmark, we decouple the evaluation of spatial reasoning in language and vision. This is important because it offers insight into the LLMs ability to ground spatial knowledge drawn from the pre-training data, without having to account for errors that may be due to failures of the vision component. In fact, recent work has shown subpar performance of state-of-the-art vision-language models on basic tasks such as object identification and mapping language utterances to visual signals (e.g.https://arxiv.org/pdf/2309.17421.pdf ; https://cdn.openai.com/papers/GPTV_System_Card.pdf; https://arxiv.org/pdf/2311.09247.pdf). Evaluating each component separately can help shed more light on what each component can and cannot do, and how they fit together. Because of the same reason, it is shown that the more recent benchmarks on similar topics have all decoupled the reasoning tasks from vision components (see the summary in our table in the general response).  In addition, in our response **Regarding Reviewer w1hA and yt2L’s concerns about the simplicity of our benchmark**, we showcased that the findings obtained from our text-only grid environments are generalizable to the more complicated cases with visual inputs.
>
> **Regarding Reviewer wB3J’s question on “why the path planning task in this benchmark can be used to measure temporal reasoning”**
>
> The path planning task in the proposed benchmark is an effective measure for temporal reasoning because it requires the model to engage in sequential thinking. The task demands that the model plans several steps ahead. This involves not only understanding the current spatial configuration but also anticipating how each move will impact future options and outcomes, in order to be able to optimize over long trajectories. In fact, a common way to represent the task formulation for our multi-goal setting is using temporal logic, a type of logic particularly designed for problems involving temporal planning (Plaku, 2016; Fainekos, 2009; Ayala, 2013 to name a few).
>
> We refer the reviewer to the following sources which similarly studied spatial-temporal reasoning under the same definition as ours; for example, [4] emphasize how AI agents interacting with a physical world need to utilize spatial knowledge to understand the positions and relations between physical objects, and temporal reasoning to assess which actions to perform over time.
>
> [1]https://ieeexplore.ieee.org/stamp/stamp.jsp?tp=&arnumber=6420843,
> [2]https://arxiv.org/abs/2103.14230,
> [3]https://books.google.com/books?hl=en&lr=&id=BIHLm7W44MoC&oi=fnd&pg=PR8&dq=spatial+temporal+reasoning&ots=-WvD0Xm1Wm&sig=Ewg2juC2BwtMy6xQ4E_EJJnnEQs#v=onepage&q=spatial%20temporal%20reasoning&f=false
> [4] https://paperswithcode.com/paper/neuro-symbolic-spatio-temporal-reasoning

---

> > ### Comment · Area_Chair_3gfx · 2023-12-03
> >
> > Do you consider Carta et al. (NeurIPS 2023) https://arxiv.org/abs/2302.02662 a missing citation and a similar prior art benchmark? It seems quite relevant.

---

### Comment · Area_Chair_3gfx · 2023-11-20
**Please engage in reviewer-author discussions**

Reviewers - I encourage you to read the authors' response carefully and let the authors know whether their response has addressed your comments.

---

### Meta-Review · Area_Chair_3gfx · 2023-12-03

**Metareview:**

This paper investigates the potential of Large Language Models (LLMs) for spatial-temporal reasoning, specifically in the area of path planning. It introduces a new benchmark, PPNL, which poses 2D grid path planning problems to LLMs and assesses their performance. The authors highlighted the limitations of the current LLMs in this domain: they find that with appropriate prompting techniques, LLMs can perform well in simple settings but struggle with long-term planning and complex environments; in contrast, while fine-tuned LLMs achieved good results on in-distribution tasks, they struggled to generalize. One main concern from the reviewers is around the contribution significance to the community. Simple benchmarks are completely fine but, in that case, people would expect such paper to provide more insights and analysis in how to advance in the target domain from a benchmark of such kind, which is the capability of spatiotemporal reasoning in this work. Overall, the reviewers think this paper is perhaps too thin in most aspects. This concern has not been fully addressed during rebuttal. Thus, the AC recommends rejecting this paper. The authors are encouraged to improve the paper based on the reviewers' comments and resubmit to a future venue.

**Justification For Why Not Higher Score:**

Lack of significant impact to the community

**Justification For Why Not Lower Score:**

N/A

---

### Decision · Program_Chairs · 2024-01-16

Reject